# CURRICULUM DISCOVERY THROUGH AN ENCOMPASSING CURRICULUM LEARNING FRAMEWORK

## ABSTRACT

We describe a curriculum learning framework capable of discovering optimal curricula in addition to performing standard curriculum learning. We show that this framework encompasses existing curriculum learning approaches such as difficulty-based data sub-sampling, data pruning, and loss re-weighting. We employ the proposed framework to address the following key questions in curriculum learning: (a) what is the best curriculum to train a given model on a given dataset? and (b) what are the characteristics of optimal curricula for different datasets and difficulty metrics? We show that our framework outperforms competing state-of-the-art curriculum learning approaches in natural language inference and two other text classification tasks. Exhaustive experiments illustrate the generalizability of the discovered curricula across the datasets and difficulty metrics.

## 1 INTRODUCTION

Curriculum Learning (CL) is a technique in Machine Learning that mimics human education systems. To learn a complex subject, students must first learn the foundational and basic materials before learning more complex ones. Without a curriculum, learning may be intractable, inefficient and learners may never reach a full understanding of a topic due to lack of the required background knowledge. Machine learning optimization through stochastic gradient descent trains models by observing example data instances. Some data instances are harder than others and require background knowledge, which could be acquired by observing and being adept at easier examples before harder ones. CL techniques seek to order examples according to their difficulty for training to generate better models. It has been shown that CL improves performance in solving harder tasks, or in cases of limited or noisy data (Wu et al., 2021). CL research has made significant progress in the last decade through the work of Bengio et al. (2009), although the principle of ordering training samples from easier to harder was introduced in the 1990s (Elman, 1993; Sanger, 1994; Rohde & Plaut, 1999).

A curriculum can be defined by ordering training samples based on their difficulty to learn. Given a measure of difficulty, there are different types of CL approaches for ordering the data: sub-sampling techniques, which sample the easiest or hardest data points at every iteration for training (Zhou et al., 2020; Bengio et al., 2009; Xu et al., 2020; Guo et al., 2018; Platanios et al., 2019), sample weighting techniques, which use the complete data at every iteration but weight data points differently according to their difficulty (Castells et al., 2020; Kumar et al., 2010; Jiang et al., 2015; 2018; Zhou et al., 2020; Yang et al., 2019), and pruning techniques, which prune the hard or noisy samples from the dataset prior to training (Northcutt et al., 2021; Guo et al., 2018). Sub-sampling methods can be cumulative, exclusive or a combination of both. Cumulative approaches add new data points to the ones that have previously been used for training. On the other hand, exclusive approaches create a new subset of data at every training stage. Such methods can introduce samples from easy to hard (Bengio et al., 2009; Kumar et al., 2010) or hard to easy, which can be an effective learning strategy in some specific tasks (Kocmi & Bojar, 2017; Zhang et al., 2018; 2019). In addition, CL methods may impose a curriculum by adjusting model's capacity according to the difficulty of their inputs (Karras et al., 2018; Sinha et al., 2020; Morerio et al., 2017) or schedule the order of tasks in the context of multi-task learning (Caubrière et al., 2019; Sarafianos et al., 2017; Florensa et al., 2017). Other CL approaches such as (Zhou et al., 2020; Saxena et al., 2019) use $\mathcal{O}(n)$, where $n$ is the number of training samples, extra parameters for learning curricula.

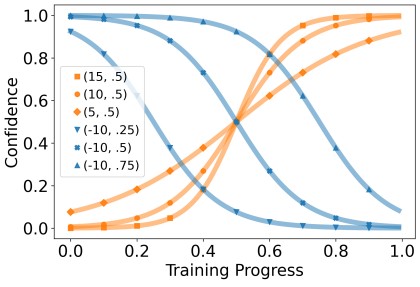
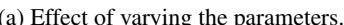

(a) Effect of varying the parameters.

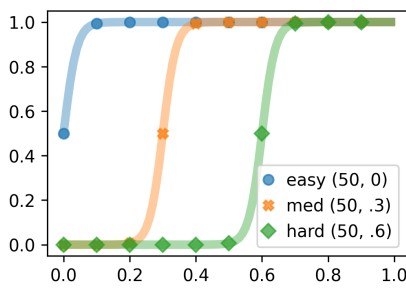

(b) Easy to Hard Curriculum.

Figure 1: The effect of the rate of growth and shift parameters $(r, s)$. (a) illustrates different weighting strategies that can be obtained by varying the rate and shift parameters. (b) a specific parameter configuration for a curriculum that first introduces easier training samples to a model, and then gradually introduces medium and hard samples at 30% and 60% of the training epochs.

Current CL approaches calculate difficulty scores for training samples based on the loss of a trained model (Xu et al., 2020; Wu et al., 2021), trainable parameters that weight samples (Kumar et al., 2010; Jiang et al., 2015; Castells et al., 2020), loss value during training (Wu et al., 2021), moving average of loss during training (Zhou et al., 2020), transformations of the loss during training (Jiang et al., 2018; Castells et al., 2020), and consistency in the correct classification of samples (Amiri et al., 2017; Xu et al., 2020). The difficulty scores will then be used to order samples for training.

In this work, we focus on an alternative approach, where examples can be ordered based on *prior knowledge* about their difficulty, e.g., object shape or orientation in image classification (Bengio et al., 2009). We will demonstrate a CL framework that encompasses existing CL approaches through an effective and flexible data partitioning and weighting scheme. Our framework provides a new paradigm for selecting an ordering strategy. Instead of a pre-determined strategy, the framework allows searching over the curriculum space to identify the best curriculum for a particular dataset and model. It partitions training data into several groups of training samples, e.g. {*easy*, *medium*, *hard*} samples, according to a difficulty scoring function. Parameterized weighting functions will then be defined for each data group to specify the weight of its samples during training. Each weight function is controlled by two parameters, which can be set empirically or adjusted using Bayesian optimization. In addition, the framework discovers optimal curricula by optimizing the parameters of the weight functions (only 2 parameters per function) using a hyper-parameter optimization algorithm. Furthermore, the curricula identified through this search provide useful insight about the dataset, such as the relative importance of different samples or knowledge dependency between samples, e.g. which samples should be learned first.

We begin by explaining our framework and showing how it is capable of approximating existing CL approaches. Then, in the context of the proposed framework, we investigate curriculum discovery, characteristics of discovered curricula, and generalizability of curricula with respect to their datasets.

## 2 CURRICULUM LEARNING FRAMEWORK

### 2.1 WEIGHTING FUNCTIONS

We define the curriculum using generalized logistic functions (Richards, 1959) of the form:

$$w(t; r, s) = \frac{1}{1 + \exp(-r * (t - s))}, \tag{1}$$

where $r \in \mathbb{R}$ is the rate-of-change parameter, which specifies how fast the weight can increase ($r > 0$) or decrease ($r < 0$); $t \in [0, 1]$ is the training progress (iteration number divided by max iterations); and $s \in \mathbb{R}$ shifts the pivot weight of the logistics function ($w(.) = .5$) to the left or right such that at $t = s$ the weight is $0.5$. Figure 1a illustrates the effect of varying these parameters. Greater absolute values for the rate parameter enforce greater slope and faster rate of changes in weights, while greater values of the shift parameter enforce longer delays in reaching the pivot

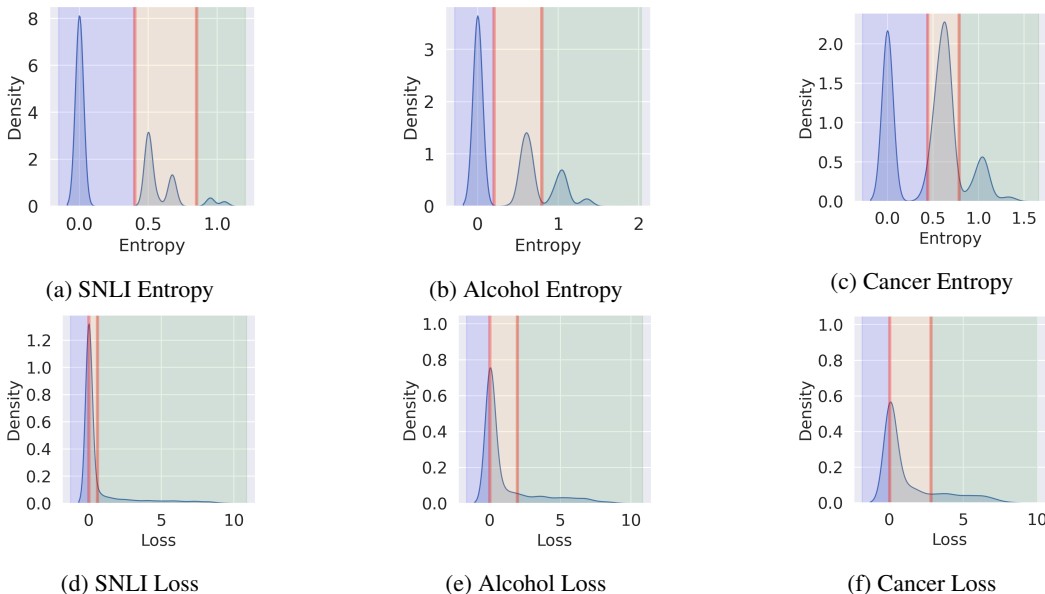

(a) SNLI Entropy      (b) Alcohol Entropy      (c) Cancer Entropy

(d) SNLI Loss      (e) Alcohol Loss      (f) Cancer Loss

Figure 2: Distributions of entropy and loss of the three datasets used in the experiments. Figures (a) - (c) show entropy, and (d) - (f) show loss. Samples of the *easy* class are to the left of the first vertical line, those of the *medium* class are between the two vertical lines, and samples of the *hard* class are to the right of the second line.

weight of $0.5$. These parameters provide flexibility in controlling sample weights during training, which is critical for deriving effective curricula. Given the weighting function, we encode prior knowledge about sample difficulty into the training paradigm of our CL framework by splitting the data into several partitions of increasing difficulty according to the prior knowledge. Then, we define a weight function by a pair of parameters $(r, s)$ for each partition,

The proposed sample weighting framework provides flexibility in sample-ordering according to difficulty. It can be learned to approximate existing predetermind curricula. For example, it is possible to begin with only easy instances or only difficult instances (as an *anti-curriculum*) or a combination of both. Figure 1b shows a specific configuration for the logistic functions based on standard CL (Bengio et al., 2009; Kumar et al., 2010), where training starts with easier samples and gradually proceeds with harder ones. In addition, our approach enables discovering new data-driven curricula from data.

Prior to training, the difficulty scores of samples are computed and each sample is assigned to a difficulty class $c \in \{$*easy, medium, hard*$\}$, see §2.2. In addition, the hyper-parameters of our three weight functions are optimized prior to training and kept fixed throughout training $\{(r_e, s_e), (r_m, s_m), (r_h, s_h)\}$. During training, the weighted loss is computed as follows.

$$\hat{l}_i = w(t; r_c, s_c) * l_i \tag{2}$$

Where $l_i$ is the unweighted and instantaneous loss of instance $i$, $\hat{l}_i$ is the weighted loss, $t$ is the current training iteration divided by the maximum number of iterations, $c$ is the difficulty class of instance $i$, and $(r_c, s_c)$ are the corresponding rate and shift parameters for the difficulty class $c$.

## 2.2 SCORING FUNCTIONS

Ground-truth labels for many datasets are often obtained through human annotation and crowd-sourcing. This is achieved by collecting multiple annotations per data sample and aggregating the results, typically by majority voting. We use sample-level annotator disagreement to define a difficulty score for each sample using Shannon entropy (Shannon, 2001), where higher disagreement among annotators corresponds to higher sample difficulty. Entropy is a natural measure of difficulty (for human population) and may serve as a reliable prior knowledge for partitioning data by

difficulty. Entropy of each sample $x_i$ is calculated as $H(x_i) = -\sum_l p_c \log p_c$ where $c$ is a class category and $p_c$ is the fraction of annotators who chose label $c$ for the sample. The use of entropy is supported by Nie et al. (2020), who studied the correlation between human agreement and model performance and reported a consistent positive correlation between model accuracy and level of human agreement, showing that model performs better on samples with a high level of agreement.

Furthermore, training loss contains valuable information about difficulty with respect to the model (learner), which may be different among architectures and tasks, and indicative of the model's specific needs. However, loss at a particular step (e.g., final loss) is dictated by the stochastic gradient descent and mini-batching dynamics and therefore is not a good indicator of difficulty (Zhou et al., 2020; Wu et al., 2021). Using a baseline model, trained with no curriculum and with default hyperparameters, we collect the loss values of all training instances at intervals of 0.5 epochs and use the average loss to estimate sample difficulty. In our experiments, we obtain twenty observations of the loss and compute the average for each instance. Such an estimation is supported by Zhou et al. (2020) who showed that the moving average of a sample's instantaneous loss is a good metric for difficulty. Partitioning the data into three groups of increasing difficulty can be done using difficulty score percentiles, or 1-dimensional k-means clustering of the scores. Examples of data partitions using entropy and loss are shown in Figure 2.

## 2.3 ENCOMPASSING FRAMEWORK

Curriculum learning approaches can be divided into three categories depending on how they process their input data: approaches that identify and prune noisy data that may hurt performance (Northcutt et al., 2021; Guo et al., 2018; Rooyen et al., 2015; Patrini et al., 2016; Chen et al., 2019), approaches that use different sub-samples of data during training (Bengio et al., 2009; Zhou et al., 2020; Xu et al., 2020; Platanios et al., 2019; Zhou & Bilmes, 2018), and approaches that re-weight loss according to sample difficulty, choosing to emphasize either easy or hard samples (Castells et al., 2020; Jiang et al., 2015; 2018; Yang et al., 2019; Saxena et al., 2019). The framework presented in this paper is capable of representing all of the three approaches.

First, data pruning can be done by assigning negative values to the rate change and shift parameters in our framework, $r$ and $s$ in Eq. 2. A negative $r$ causes the weight to approach zero, and a negative $s$ shifts the curve to the left, so the curve reaches zero before training begins. The framework also allows flexibility in pruning: by setting a small positive $s$, the noisy data can be seen by the model for a short amount of time before reaching zero weight, or by setting a positive $r$ and a large positive $s$ the noisy data will only be seen at the end of training (after it stabilizes).

Second, data sub-sampling can be represented by the weight going to zero or increasing from zero at different stages of training. For instance, Figure 1b illustrates a curriculum where the easy samples are sub-sampled in the beginning, and harder samples are introduced at later stages.

Third, we display in Figure 3 the confidence scores assigned to our data by three loss re-weighting approaches. The results are generated by our implementations of the three approaches, evaluated on the three datasets introduced in §3.1, where each model runs with five random seeds. The partitioning of *easy*, *medium*, and *hard* is according to the entropy-based difficulty classes, as described in §2.2. We record the average weight (confidence) assigned to each class. The result is averaged over all runs, and the shaded area is the 95% confidence interval. The approaches that estimate sample confidence based on loss (Castells et al., 2020; Zhou et al., 2020; Kumar et al., 2010; Jiang et al., 2015; Felzenszwalb et al., 2009) tend to generate monotonic curves over the course of training because training loss tends to be non-increasing at every step. Therefore, the confidence scores assigned by these re-weighting approaches follow a monotonic curve that can be approximated by our weighting functions (§2.1, Figure 1). We note that although the weight scale of SuperLoss (Castells et al., 2020) in Figure 3a is larger than one, this model can still be represented by our CL framework because the increased scale corresponds to a scaling of the learning rate, as shown below:

$$\theta_t = \theta_{t-1} - \eta \nabla \frac{1}{n} \sum_i \sigma_i l_i = \theta_{t-1} - (\eta \cdot \sigma_{max}) \nabla \frac{1}{n} \sum_i \frac{\sigma_i}{\sigma_{max}} l_i, \qquad (3)$$

where $l_i$ and $\sigma_i$ are the loss and confidence of sample $i$, respectively. Therefore, our framework can also represent CL approaches with a confidence scale larger then one.

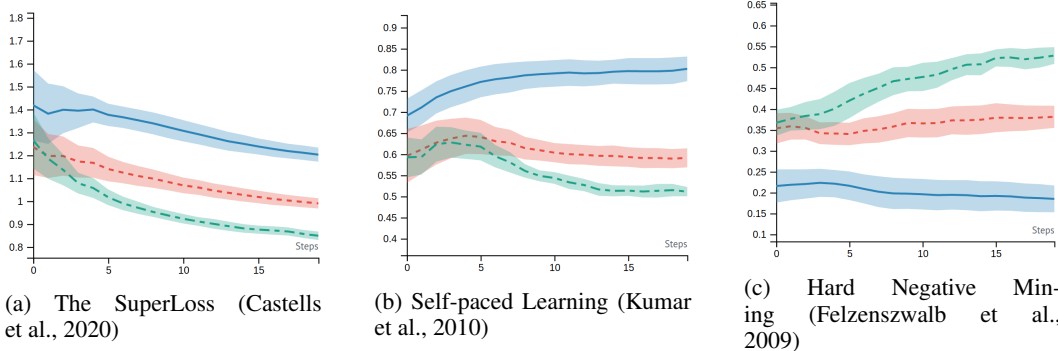

(a) The SuperLoss (Castells et al., 2020)

(b) Self-paced Learning (Kumar et al., 2010)

(c) Hard Negative Mining (Felzenszwalb et al., 2009)

Figure 3: Confidence assignment to samples in our datasets by three curriculum learning approaches that re-weight their loss functions by computing confidence scores for samples. The x-axis is the epoch number, and y-axis is the average weight assigned to instances of different difficulty. Blue (solid) is *easy*, orange (dashed) is *medium*, and green (dash-dot) is *hard*. The shaded area is the 95% confidence interval (CI) over three datasets with five random seeds each. The curves are monotonic for most parts, and can be approximated by the monotonic curves generated by our framework.

## 2.4 CURRICULUM DISCOVERY

We employ a hyperparameter optimization framework (Akiba et al., 2019) to find the optimal value of the curriculum parameters $(r, s)$ for each difficulty class. Using this method, we can learn a data-driven curriculum beyond what we could manually design through empirical settings or a choice among the limited ordering strategies. To optimize the three pairs of parameters, we use the Tree-structured Parzen Estimator (TPE) sampling algorithm (Bergstra et al., 2011). Unlike grid or random search (Bergstra & Bengio, 2012), TPE traverses the parameter space by estimating the parameters that are most probable to perform better on a trial, based on the previous trials. TPE defines two Gaussian Mixture Models, $l(x)$ and $g(x)$ which are formed using the best and remaining observed parameters, respectively. TPE selects the parameter $x$ with a high probability under $l(x)$ and low probability under $g(x)$, i.e. $\arg\max_x l(x)/g(x)$. This choice of sampling algorithm greatly speeds up the search of our curriculum parameters.

We note that the discovered curricula are optimal within this framework, constrained by the method of data partitioning and the class of weight functions. We argue that the proposed framework is able to approximate curricula defined by existing CL approaches, and outperform existing CL approaches across several datasets.

## 3 EXPERIMENTS

### 3.1 DATASETS

We evaluate our approach on three datasets that contain multiple annotations for each sample. First, the Stanford Natural Language Inference (SNLI) benchmark (Bowman et al., 2015), which contains 550k training samples, 10k development samples, and 10k test samples. Within the training samples, there are 36.7k samples annotated by 5 workers and 2.6k annotated by 4 workers, which we use for our experiments and refer to as SNLI "full." Furthermore, in order to control for variance due to imbalanced difficulty classes, we create a balanced subset of the data. As shown in Figure 2a (notice the y-axis scale), SNLI is highly imbalanced in entropy classes. The data is downsampled by selecting an equal number of samples from each entropy-class. The balanced subset contains a total of 2.3k samples, i.e. 774 samples in each entropy class. On the other hand, loss classes are fairly distributed, see Figure 2d. The downsampled subset contains both entropy and loss classes which will be used in experiments.

The Alcohol dataset (Amiri et al., 2018; Weitzman et al., 2020) has been developed to obtain population-level statistics of alcohol use reports through social media. The dataset consists of more than 9k tweets. Given an alcohol relevant tweet, annotators are asked to determine if it reports first-

person alcohol use, and if yes, the intensity of the drinking (light vs. heavy), the context of drinking (social vs. individual), and the time of drinking (past, present, or future). All samples in the dataset are labeled by at least three workers, including over 1.3k samples labeled by five or more workers. We define a multi-class classification task for this dataset based on alcohol relevance, intensity and context of drinking. The categories and their data distributions are reported in Appendix A. We randomly split the data into 5.4k training samples, 1.8k development samples, and 1.8k test samples. The balanced version of the training set contains a total of 2.5k training samples, i.e. 863 samples in each entropy class.

The Cancer dataset has been developed to obtain population-level statistics of cancer patients; it contains 3.8k Reddit posts. Annotators are asked to determine if the post describes the experience of a cancer patient, the type of cancer, and the relation of the author of the post to the patient. We define a multi-class classification task based on post relevance and caner type. The categories and their data distribution are reported in Appendix A. All samples are labeled by at least three workers, including about 1k labeled by at least five. We randomly split the data into around 2.2k training samples, 765 development samples, and 765 test samples. The balanced version of the training set contains a total number of 1.7k sample, i.e. 578 samples in each entropy class.

We note that the datasets are significantly different in average document length, ranging from 10 (SNLI), to 15 (Alcohol) to 174 (Cancer) words. This variation can induce significant variance in the created models.

## 3.2 BASELINES

We compare the performance of our CL approach with the following state-of-the-art approaches. SuperLoss (SL) (Castells et al., 2020) is a CL approach that defines a task-agnostic confidence-aware loss function. It infers the confidences of instances from the instance loss with minimal cost, providing a closed-form solution function for estimating confidence. It up-weights samples with smaller loss values (easier instances), while down-weights those with larger loss values (harder ones) (Figure 3a). MentorNet (Jiang et al., 2018) uses an auxiliary network to generate a weight for training samples at every training iteration. It incorporates additional signals such as epoch number and instance loss history to learn data-driven curricula and is particularly strong against noisy data. Difficulty Prediction (DP) (Yang et al., 2019) defines a difficulty score based on multi-annotator data, and weight samples according to the following formula $w = 1 - \alpha(d_i - \tau_{DP})/(1 - \tau_{DP})$, where $d_i$ is the sample difficulty and $\tau$ is a pre-defined threshold.

As discussed before, our approach employs two scoring functions (§ 2.2) and two curriculum configurations for each dataset. A curriculum configuration refers to a particular setting of the six parameters controlling the three weight curves (§ 2.1). The two scoring functions are labeled as *Loss* and *Ent* (entropy). In addition, the first curriculum configuration is a gradually increasing approach in Figure 1b named *inc.*, this configuration is applied identically to all models. The second configuration is the specialized configuration (*sp.*) that is obtained through hyper-parameter search as discussed in § 2.4.

## 3.3 SETTINGS

We tune the parameters $\lambda$ of SL and $\alpha$ and $\tau_{DP}$ of DP using development data. The optimal values found are $\lambda = 1.2$, $\alpha = 0.9$ and $\tau_{DP}$ is set dynamically upon loading the dataset to the 50-percentile difficulty value. Following Castells et al. (2020), we set $\tau_{SL}$ (instances with $l_i > \tau_{SL}$ are considered hard) to the moving average of the loss in all experiments.

We use the transformers python package (Wolf et al., 2020), using the `bert-base-uncased` model for SNLI and Cancer, and `twitter-roberta-base` for Alcohol. We set learning_rate to $1e-5$, batch_size to 16, epochs to 10 (we confirm that this number of iterations is sufficient for all models to converge), and the optimizer to Adam (Kingma & Ba, 2017). For each experiment, we train five models using five random seeds applied to both `pytorch` and `numpy`. Additionally, during all data pre-processing, splitting, and sub-sampling, the random seed is set to 0, and a single NVIDIA A100 40GB GPU is used for training. The development set is used to determine the best training step which is used for the final evaluation.

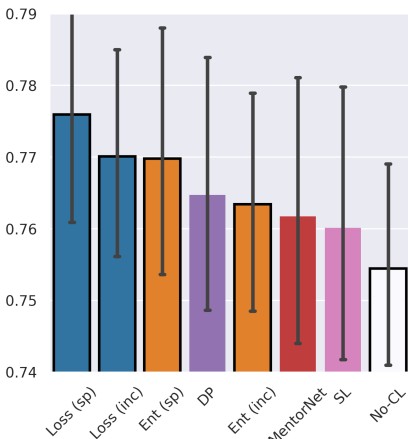

Figure 4: Overall accuracy averaged over three datasets with five random seeds. *Loss* and *Ent* indicate curricula that partition the data based on difficulty classes determined by loss and entropy respectively (§2.2). *inc* is the easy to hard curriculum shown in Figure 1b, *sp* is the specialized curriculum obtained by curriculum discovery (§2.4), which is different for each dataset.

In addition, we conduct a hyper-parameter search over the $(r, s)$ for each weight function or difficulty classes (easy, medium, and hard). We set the search space of $r$ to be from $-10$ to $10$ with a step of $2$, and of $s$ to be $-0.5$ to $1.5$ with a step of $0.25$. We observe that changes smaller than this step size have little effect on performance. This search space consists of 11 possible values for $r$ and 9 for $s$, for a total of 970k combinations. The search is run for at least 100 trials, as compared to over 1000 trials by random search, using the method described in §2.4. Each trial is run with 5 random seeds and the result is averaged. The search objective is to maximize accuracy over development data.

## 3.4 Performance Gain from Curricula Discovery

Results are shown in Figure 4. Accuracy is averaged over the six datasets (full and balanced version of each dataset). The gradually increasing curriculum (*inc*) achieves a significant improvement over *No-CL* using either of the scoring functions while being a static, off-the-shelf curriculum configuration. This improvement shows the effectiveness of our approach of partitioning the data using generalized logistic functions (§2.1). Moreover, both (*inc*) and the specialized (*sp*) curricula obtained through curriculum discovery perform significantly better than the state-of-the-art CL approaches. In our three datasets, loss as a scoring function performs better than entropy on average. This is expected as loss values can capture sample difficulty with respect to the downstream learner (model), as apposed to entropy values which do not take into account the model.

Appendix B includes further breakdown of the results by dataset and accuracy across samples of different difficulty.

## 3.5 Characteristics of Discovered Curricula

Figure 5 shows the mean and 95% CI of the top 25 performing configurations on our datasets and scoring functions. We observe several insightful patterns: the resulting curricula are non-trivial and greatly differ from the known strategies reported in current literature, such as gradually increasing difficulty or anti-curriculum. In addition, the weights of hard samples tend to approach zero, supporting the hypothesis that either these instances are too difficult for the models to learn or they are noisy. The results support the principle of pruning techniques because noisy samples induce more noise than useful signal; in several plots, the weight of hard samples increases only at the end of training after the model stabilizes. In addition, in SNLI and Alcohol *easy* samples carry the most significant weight, unlike Cancer, where *easy* samples are down-weighted early during the training. These weighting patterns reveal the relative importance of samples in each dataset. Finally, the full SNLI dataset with entropy-class partitions provides useful information. Figure 2a shows that entropy classes are highly imbalanced, with *hard* samples being much fewer than *easy* ones. In the

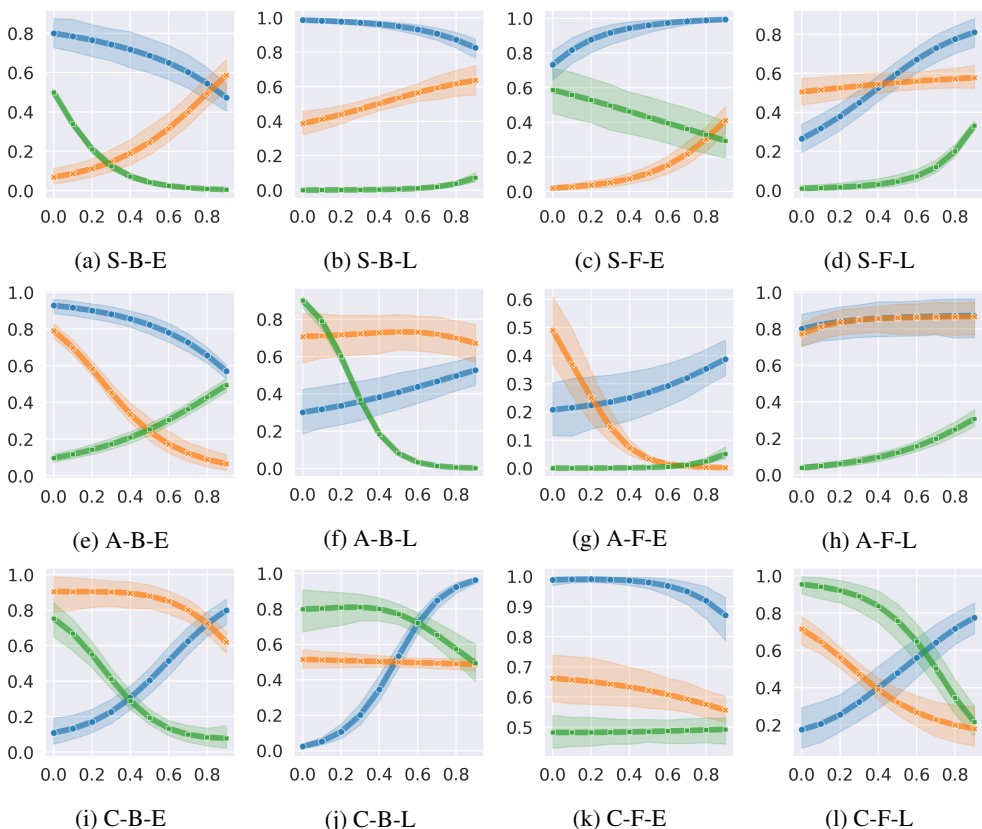

Figure 5: Each caption is composed of the first character of the name of a dataset: {**S**NLI, **A**lcohol, **C**ancer}, the type of the dataset {**B**alanced or **F**ull}, and the difficulty score used {**E**ntropy, **L**oss}. The x-axis is the training progress and y-axis is the confidence assigned to samples of a difficulty-class. The blue line (circle marker) is *easy*, orange line (x marker) is *medium*, and green line (diamond marker) is *hard*. The solid line is the mean of the top 25 performing configurations for each dataset and scoring function pair, and the shaded area represents the 95% CI.

optimal curriculum shown in Figure 5c, *hard* samples are assigned weights around 0.5, unlike the three other cases of SNLI. We attribute this result to the reduced presence/effect of *hard* samples.

## 3.6 GENERALIZABLE CURRICULA

Figure 6 shows the accuracy obtained when training using different curriculum configurations from Figure 5. Each cell in the figure is the average result of 5 seeds. We observe common characteristics among datasets that cause the curriculum to be transferable between them. First, the top three configurations are all products of down-sampled, balanced datasets. Second, the curricula obtained using the small balanced datasets generalize and achieve high performance on the large datasets. This is useful as it allows performing the hyper-parameter search much faster and cheaper on smaller datasets, providing evidence that the framework can be applied to large datasets by searching for a curriculum on a small subset of the data. Third, the *inc* curriculum is available off-the-shelf and performs consistently well across datasets and scoring functions. Fourth, as noted previously, instances of the Cancer dataset consist of long paragraphs, causing high variance in models trained using the dataset. Consequently, the curricula obtained using the Cancer and loss as measure of difficulty are of lower quality and perform the worst.

An extended version of Figure 6 is included in Appendix C with results of models trained with balanced versions of datasets.

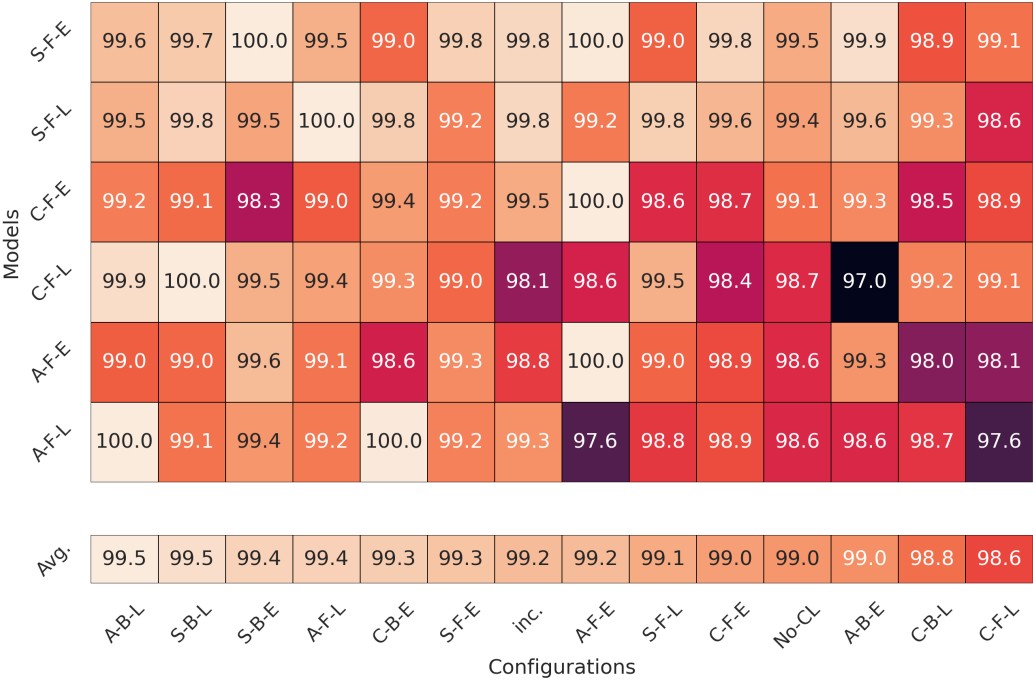

Figure 6: Using the same notation as Figure 5, the x-axis lists different curriculum configurations from the increasing curriculum *inc* (Figure 1b), to *No-CL*, to curricula discovered using a particular dataset and scoring function. The y-axis lists models that are trained using each configuration. For example, the cell at the intersection of row "S-F-L" and column "A-F-E" represents a model trained on SNLI full dataset that is partitioned by loss as a measure of difficulty, using the curriculum discovered for the full Alcohol dataset partitioned by entropy (Figure 5g). Each row of the table is normalized to match the scales of the different models.

## 4   CONCLUSION AND FUTURE WORK

We introduce an effective curriculum learning (CL) framework that employs prior knowledge about sample-level difficulty in its training paradigm, and effectively creates and explores a curricula space for curricula discovery and model generalizability. The proposed framework partitions its input data into three groups of increasing difficulty, defines three parameterized logistic weight functions to weight the loss of samples in each of the three groups, and provides the capability of tuning the parameters of the weight function to discover new curricula performing better than existing baselines. We demonstrate that this framework is capable of representing major CL approaches. In addition, an important advantage of our approach is that the curricula that it discovers for smaller and balanced datasets work well on larger datasets, across the three datasets that we experimented with. The proposed research opens a new paradigm in CL by removing the limitations imposed by selecting a single CL strategy, and instead, using a flexible framework to discover optimal curricula for given datasets and models, and extract valuable insights about the data. Our work has the limitation of using monotonic logistic functions because the weight curves can not take arbitrary forms. This challenge can be addressed in future work by adding an extra set of logistic functions and having every weight function be a linear combination of two or more logistic functions, which makes it possible to represent non-monotonic functions. Nevertheless, we have demonstrated that major CL approaches that estimate difficulty as a function of training loss result in monotonic curves because of gradient descent that causes training loss to be non-increasing.

There are several promising areas for future work. These include approaches for learning new difficulty indicators that are centered on data (e.g., text difficulty), prioritizing medium level instances and those with greatest progress in training, and developing challenge datasets that contain diverse data samples with different level of difficulty.

## 5 ETHICS STATEMENT

This investigation included publicly available yet sensitive data from social media, developed for an important task: obtaining population-level statistics about different public health issues. Our work does not access, use, or release any personal data, and only textual data is used for experiments.

## 6 REPRODUCIBILITY STATEMENT

The source code is included as supplementary material. It provides the required details for data processing, hyper-parameter setting, random seed setting, model evaluation, etc.

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

# A DATA CATEGORIES DISTRIBUTION

| Class | Count |
|---|---|
| (no) | 5,325 |
| (yes, light use, individual) | 1,464 |
| (yes, heavy use, individual) | 964 |
| (yes, not sure, individual) | 457 |
| (yes, heavy use, other) | 423 |
| (yes, heavy use, group) | 284 |
| (yes, light use, group) | 161 |
| **Total** | 9,078 |

(a) Alcohol

| Class | Count |
|---|---|
| (irrelevant, no patient experience) | 1,996 |
| (relevant, breast cancer) | 617 |
| (relevant, colon cancer) | 444 |
| (relevant, brain cancer) | 284 |
| (irrelevant, none of the above) | 251 |
| (irrelevant, other cancer types) | 162 |
| (irrelevant, news related to cancer) | 70 |
| **Total** | 3,824 |

(b) Cancer

Table 1: Statistics of the Alcohol and Cancer datasets.

# B  ACCURACY BREAKDOWN

The accuracy achieved on each dataset using each approach is shown in Figure 7.

Models that achieve a high accuracy may be generating correct predictions for a high percentage of easy samples while failing to correctly predict the output of the *medium* and *hard* samples. Standard evaluation benchmarks often contains artifacts that make it east to correctly predict the label of some samples (Gururangan et al., 2018; Poliak et al., 2018). Therefore, it is important to closely analyze the model's performance on harder instances. We break down the accuracy of each difficulty class (with entropy-class partitioning) for balanced, full, and all datasets in Figures 8, 9, and 10, respectively. Our approach is exceptionally powerful in predicting samples of *medium* difficulty. Furthermore, the accuracy achieved on *hard* samples is almost equal by all approaches, including No-CL, supporting the hypothesis that those samples tend to be noisy or inaccurately labeled.

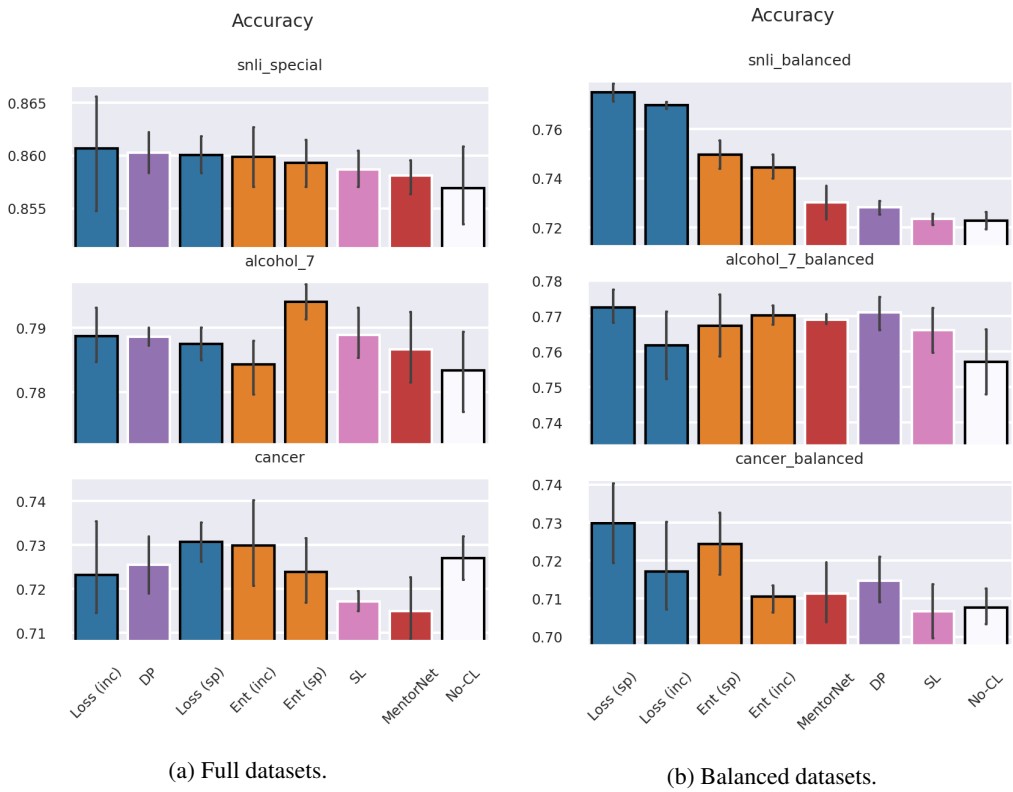

(a) Full datasets.

(b) Balanced datasets.

Figure 7: Accuracy of different CL approaches on each dataset.

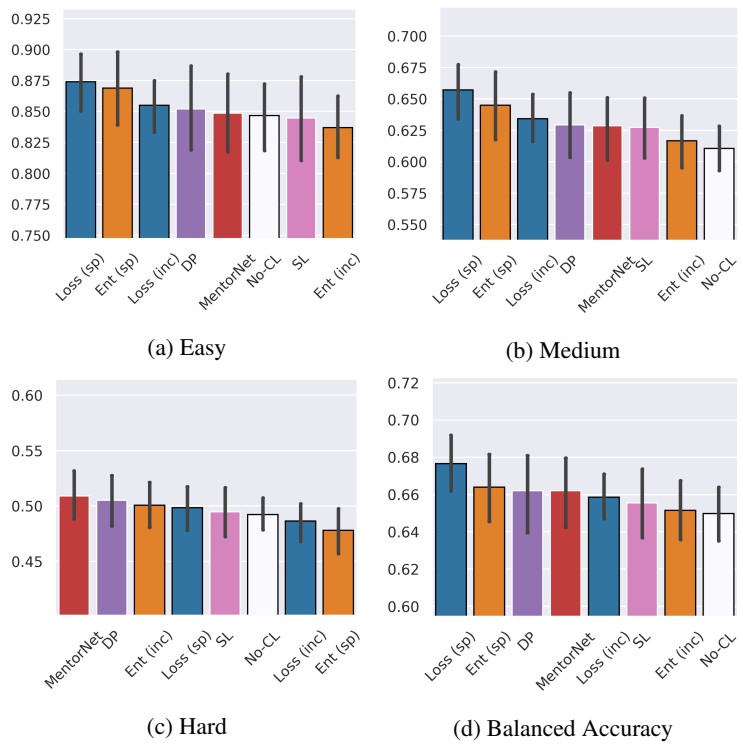

(a) Easy      (b) Medium

(c) Hard      (d) Balanced Accuracy

Figure 8: Average over balanced datasets.

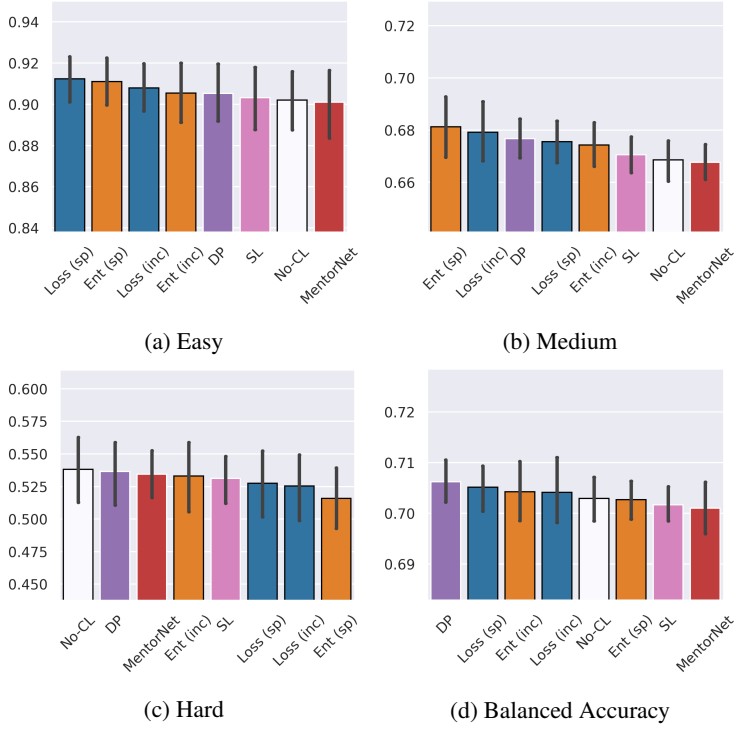

(a) Easy      (b) Medium

(c) Hard      (d) Balanced Accuracy

Figure 9: Average over full datasets.

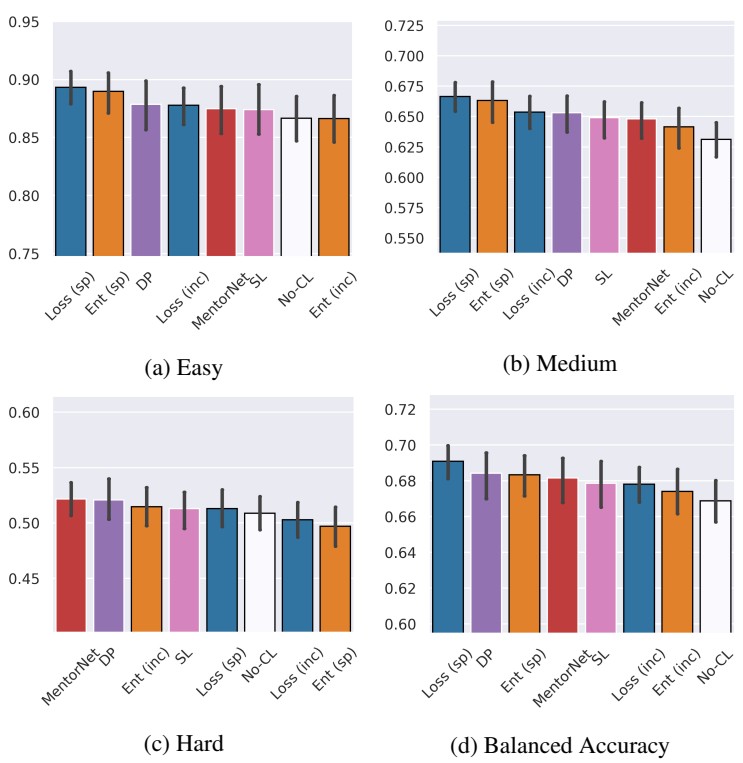

(a) Easy

(b) Medium

(c) Hard

(d) Balanced Accuracy

Figure 10: Average over all datasets.

# C EXTENDED CONFIGURATION GENERALIZABLITY EXPERIMENTS

Figure 11: An extended version of Figure 6 including experiments on balanced versions of the datasets.

Figure 11 shows the full results including evaluation on the balanced datasets.

# D Full Curriculum Search Results

This section shows the top 25 performing configurations on each dataset scoring function pair. Blue lines (circle marker) is *easy*, orange (x marker) is *medium*, and green (diamond marker) is *hard*. Above each plot is the development set accuracy. The plot legend contains the parameters $(r, s)$ for each class.

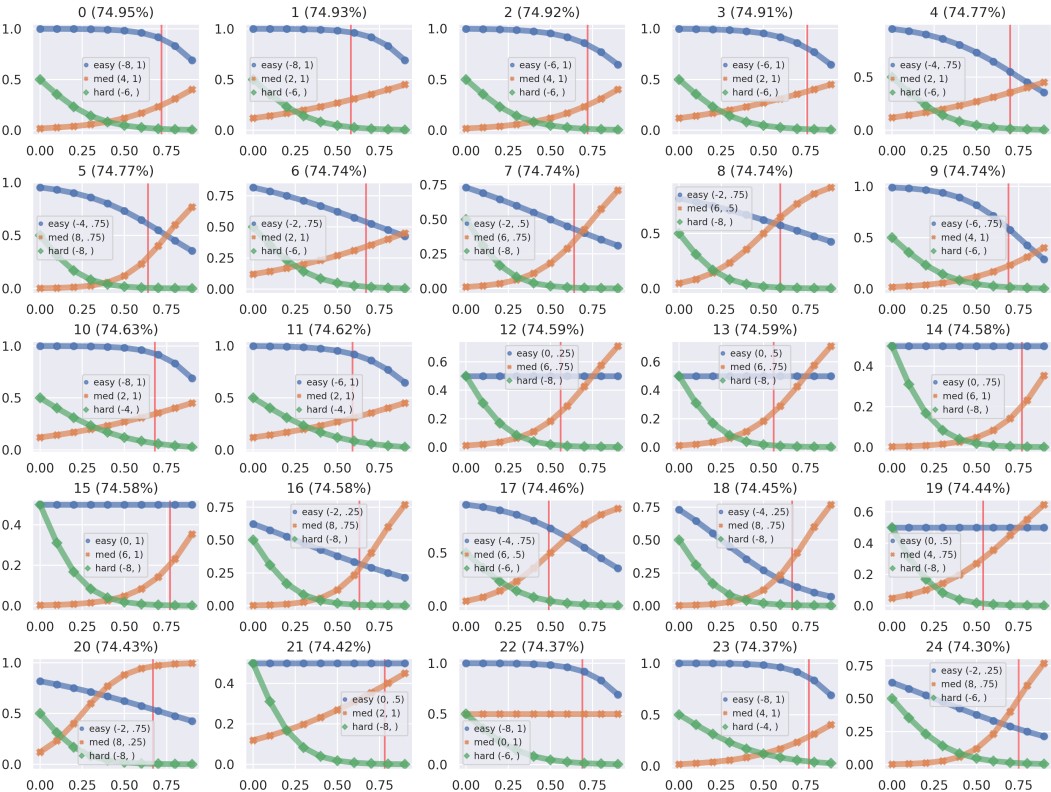

Figure 12: S-B-E

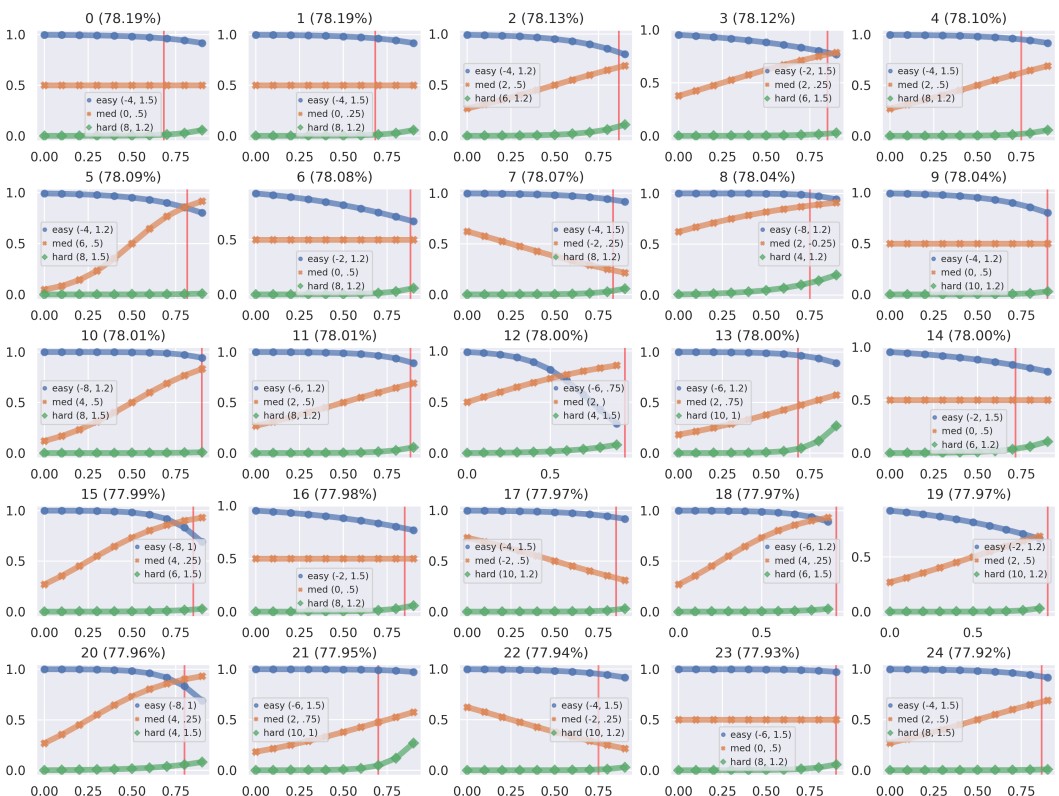

Figure 13: S-B-L

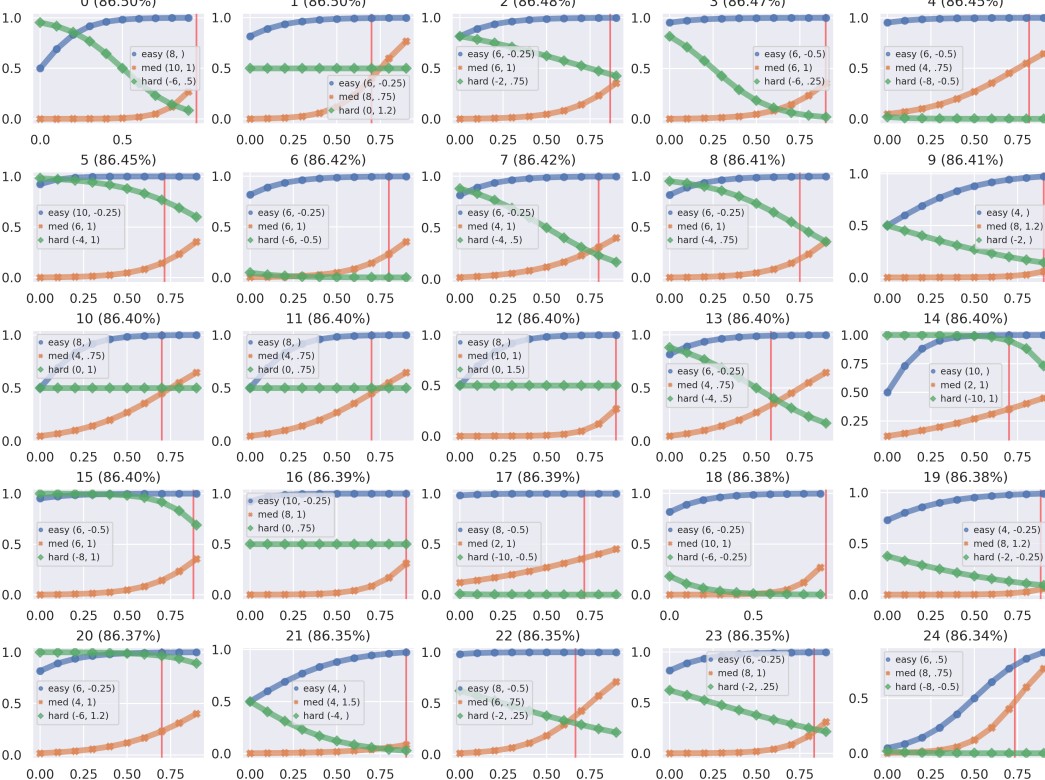

Figure 14: S-F-E

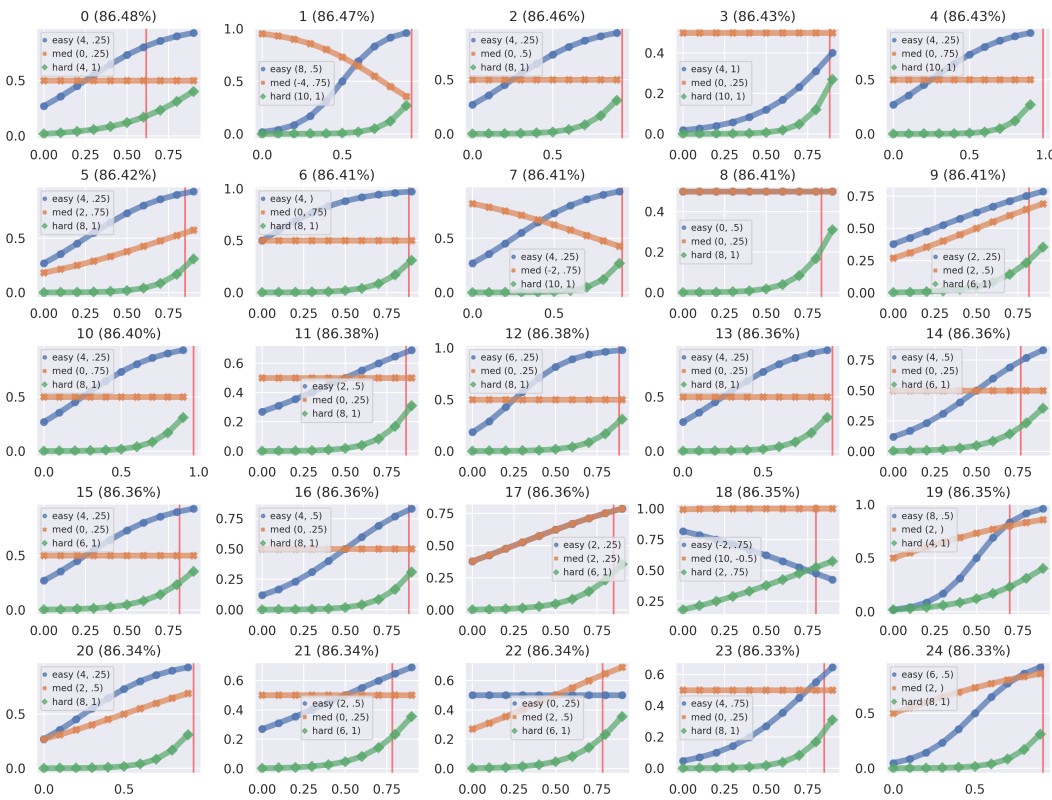

Figure 15: S-F-L

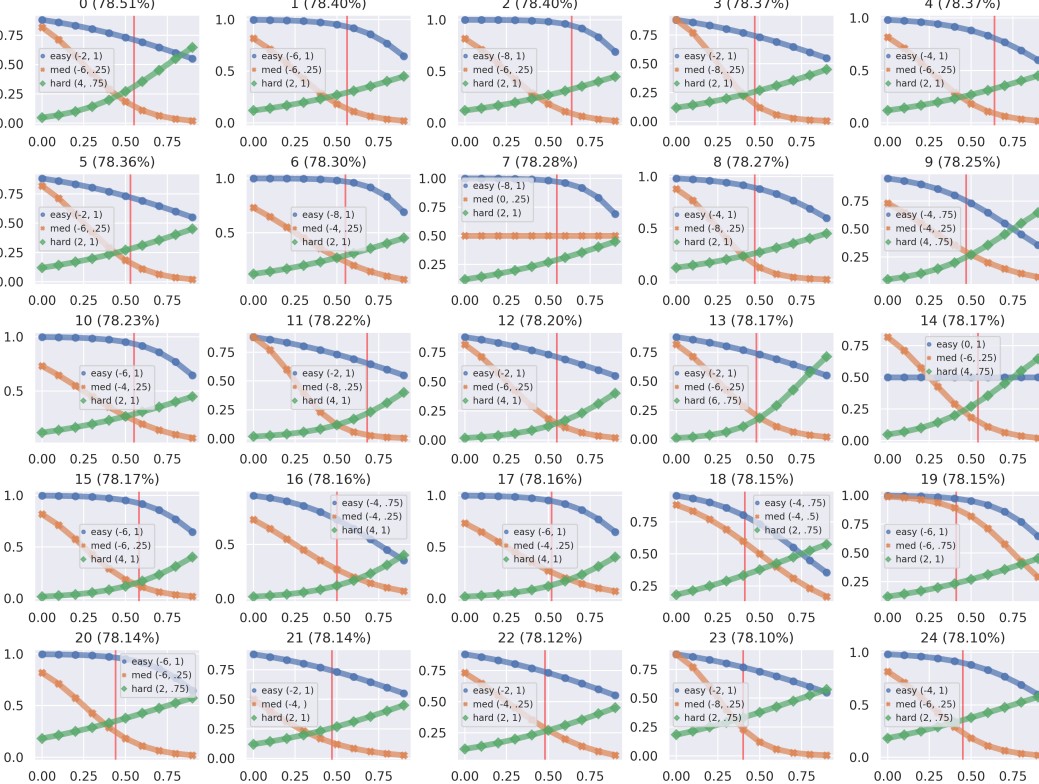

Figure 16: A-B-E

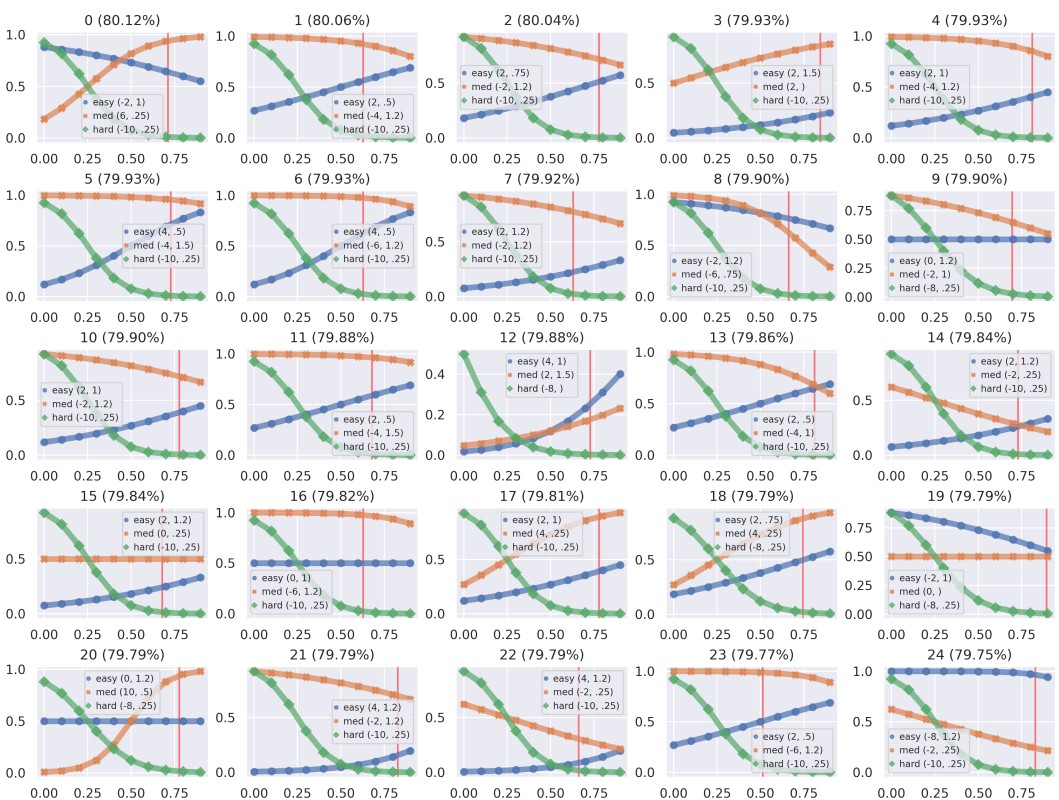

Figure 17: A-B-L

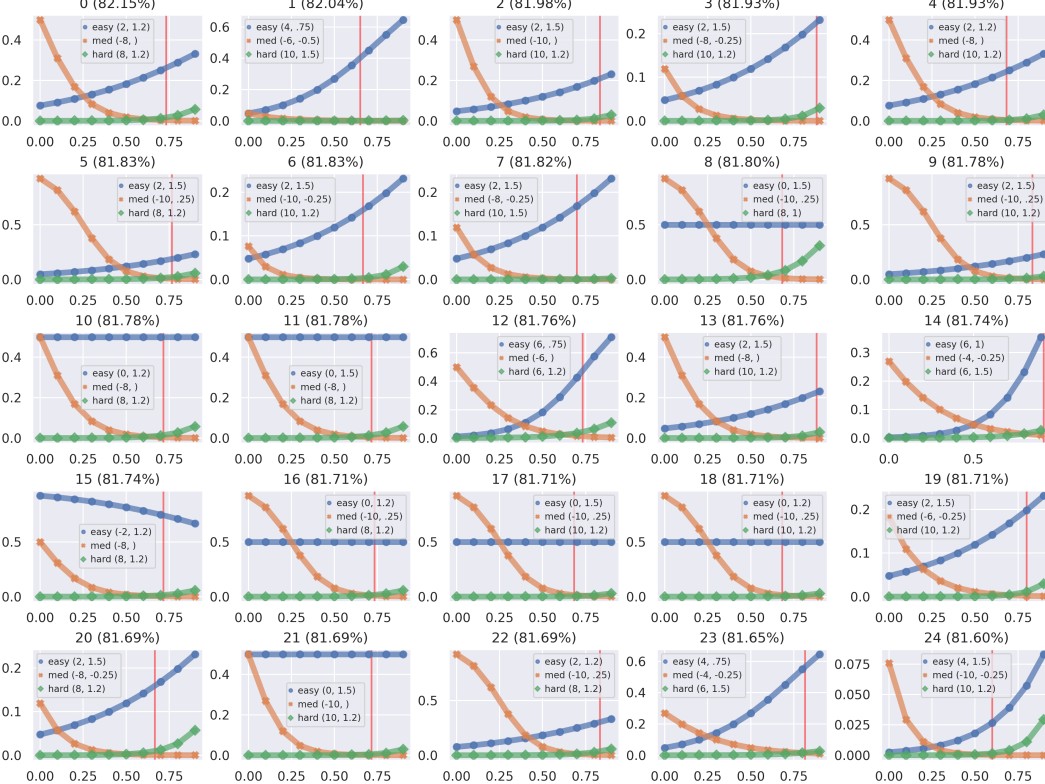

Figure 18: A-F-E

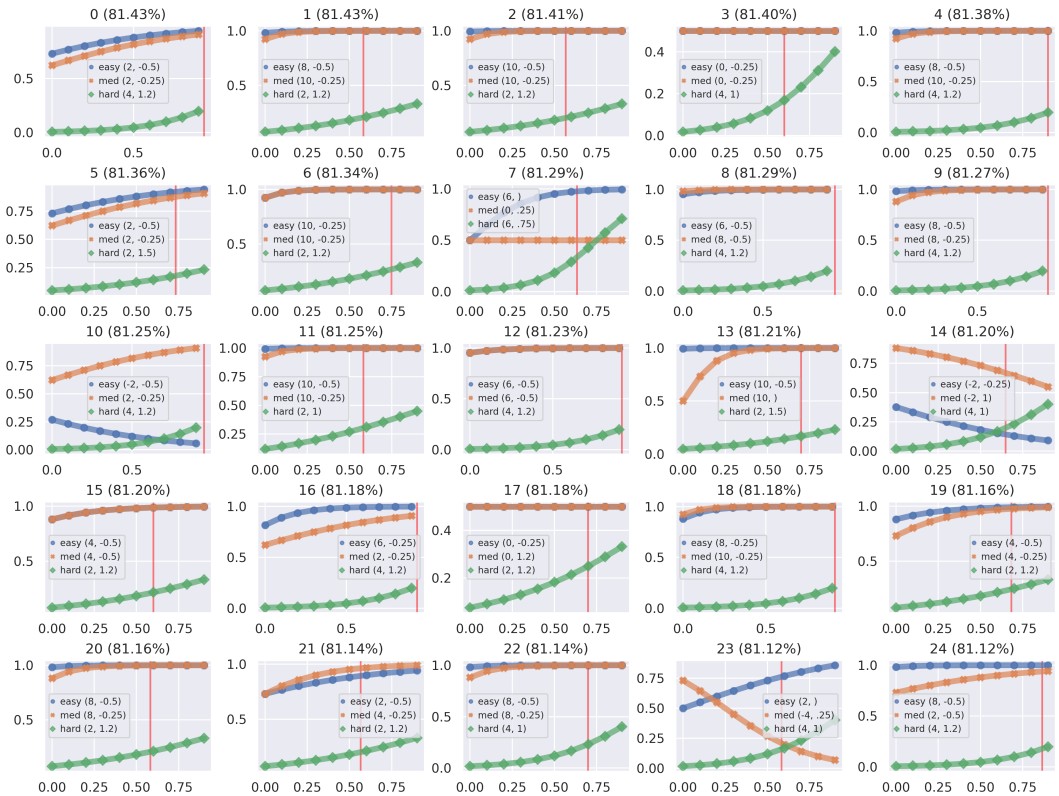

Figure 19: A-F-L

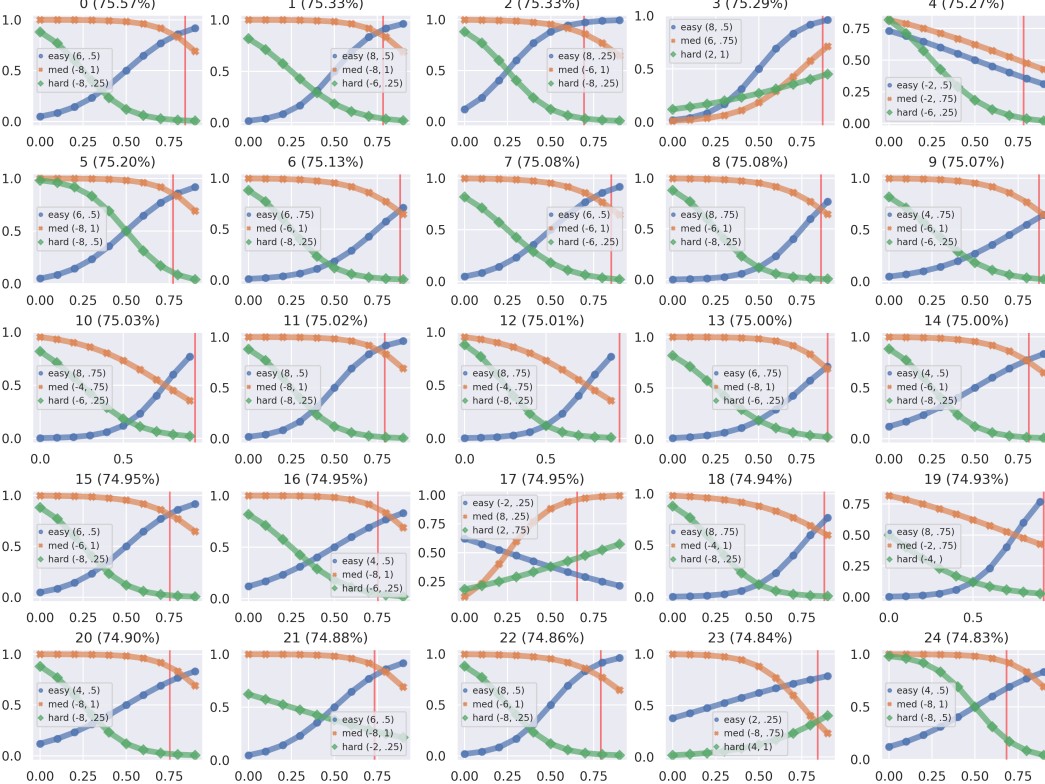

Figure 20: C-B-E

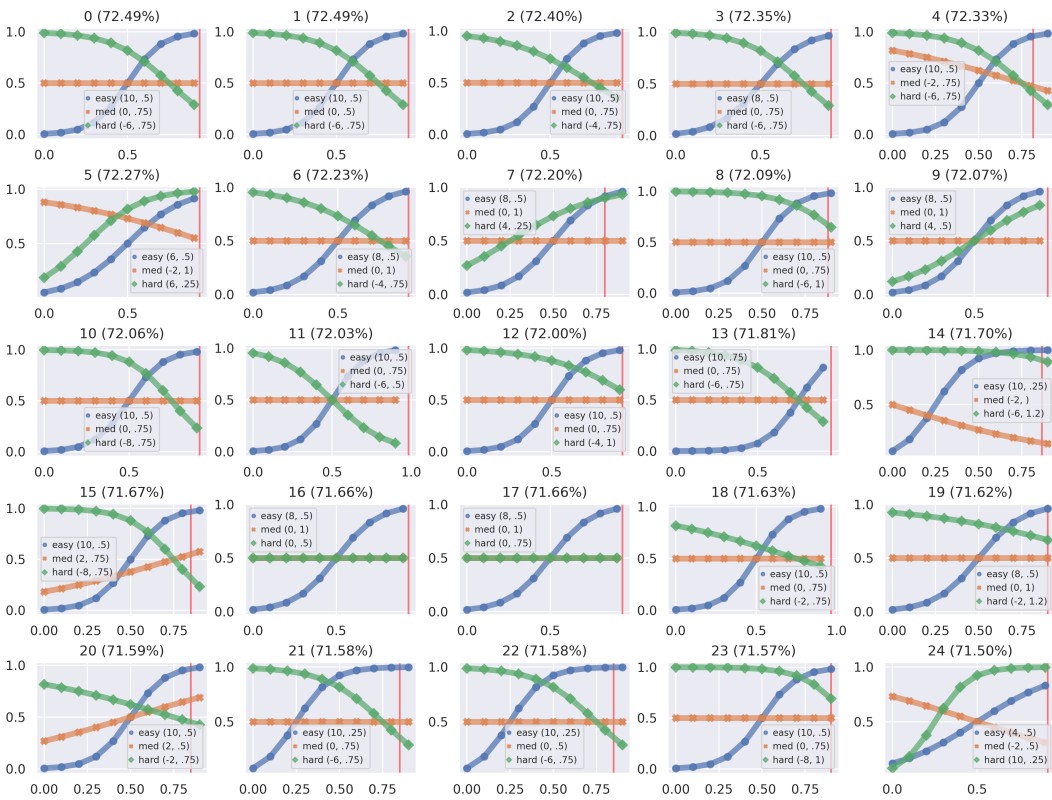

Figure 21: C-B-L

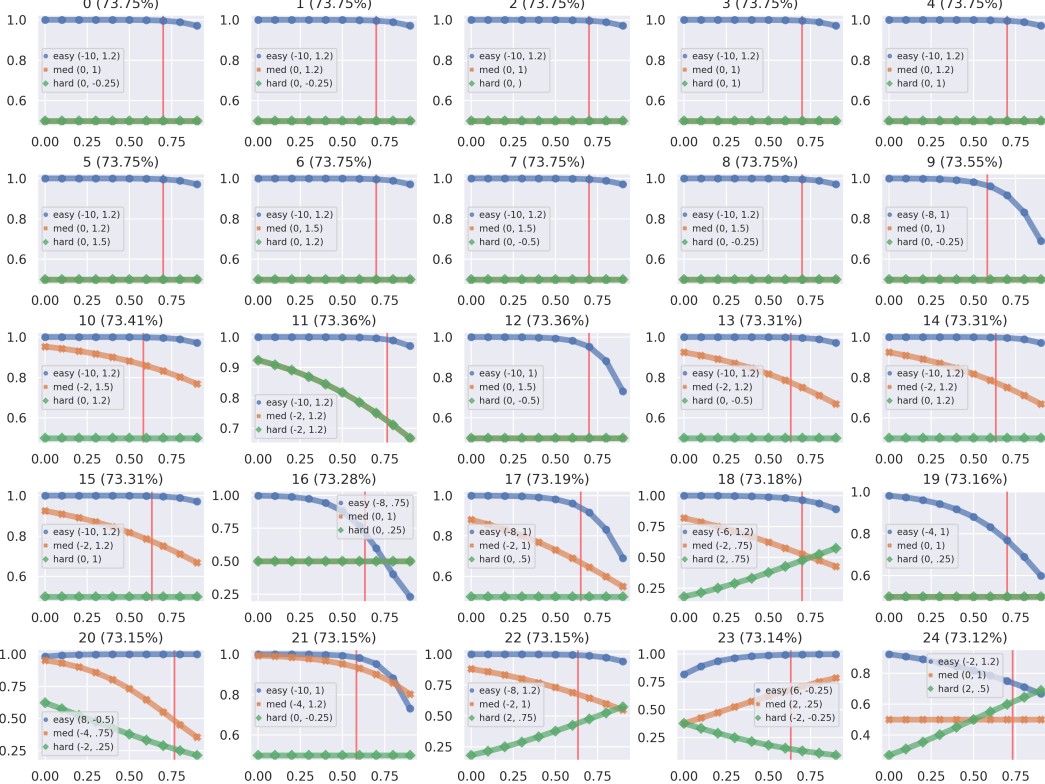

Figure 22: C-F-E

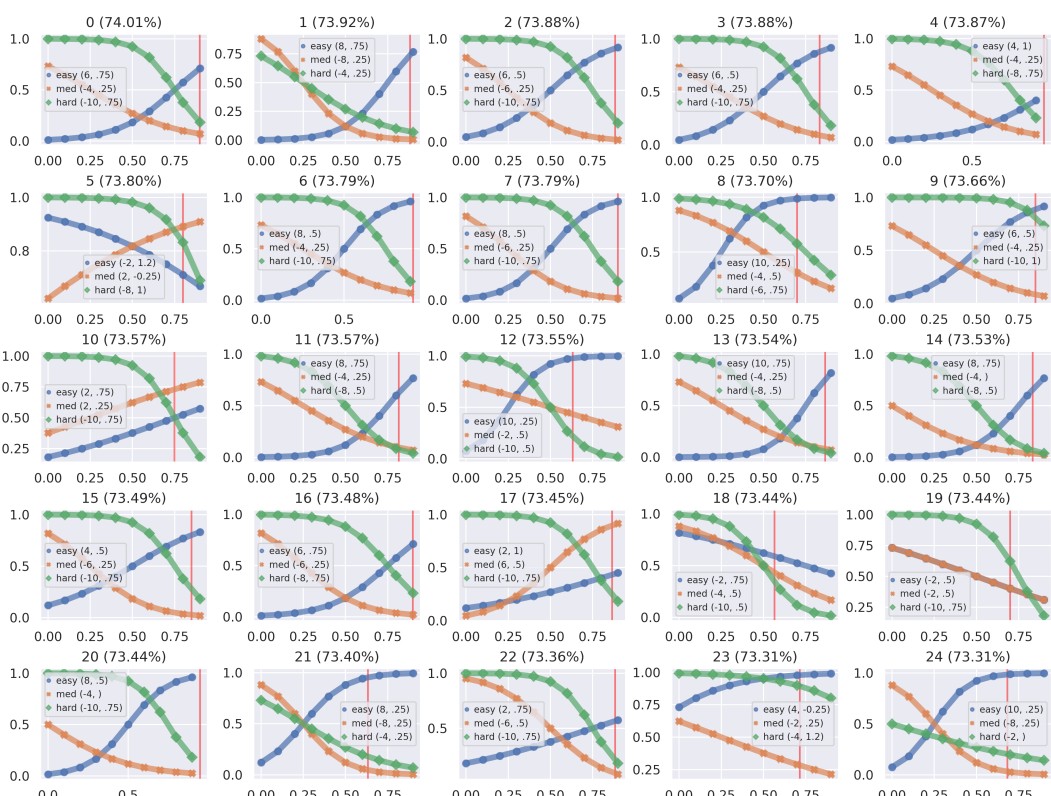

Figure 23: C-F-L

