# OpenReview forum: "Curriculum Discovery through an Encompassing Curriculum Learning Framework"
_ICLR.cc/2022/Conference — ICLR 2022 Submitted_

### Official Review · Reviewer_mHjR · 2021-10-17

**Correctness:** 2
**Technical Novelty And Significance:** 2
**Empirical Novelty And Significance:** 3
**Recommendation:** 5
**Confidence:** 4

**Main Review:**

Strengths:
+ The authors conducted a braod set of experiments.
+ The paper is easy to follow.

Weaknesses:
- The main contribution of this work is to add changing weights to easy, medium, and hard training subsets, which an incremental development in my opinion.

- At the beginning of section 2.3, the authors should acknowledge curriculum methods that do not maipulate data samples, such the "model-level" curriculum approaches surveyed in [A].

- The approach divides the training data into 3 parts {easy, medium, hard}, based on a difficulty score. Ablation results with a different number of parts and other difficulty scores should be presented. I think it is particulary interesting to see what happens if th examples are randomly partitioned, i.e. does the difficulty play an important role?

- The introduction leaves the impression that weights for data shards are somehow predicted by the model at each iteration. However, in section 2.4, the reader learns that "curriculum discovery" is based on hyperparameter tuning. The claims from the introduction should be toned down.

- The performance gains shown in Figure 4 are rather small (wintin the standard deviations) compared to competing curriculum methods. The same applies to the supplementary.

Formatting and language mistakes:
- "Without a curriculum learning will be an intractable" => "Without a curriculum, learning will be an intractable";
- "Nie et al. (2020), which also studies" => " Nie et al. (2020), who also study";
- many citations are wrongly formatted, e.g. "logistic functions Richards (1959)" should be "logistic functions (Richards, 1959)".

References:
[A] Petru Soviany, Radu Tudor Ionescu, Paolo Rota, and Nicu Sebe. Curriculum learning: A survey. arXiv preprint arXiv:2101.10382, 2021.

**Summary Of The Paper:**

The paper presents a curriculum learning approach applied to NLP models. Texts are separated into easy, medium and hard subsets based on a difficulty score that is given a standard NLP model. Hyperparameter tuning is used to determining evolving weights for each batch. The approach is evaluated on three benchmark datasets (SNLI, Alcohol, Cancer).

**Summary Of The Review:**

In my opinion, the weaknesses outweigh the strengths of this paper.

---

> ### Author Response · Authors · 2021-11-15
> **Authors' Response to Reviewer mHjR**
>
> We thank the reviewer for their time and comments. The following is a response to comments raised by the reviewer.
>
> **The main contribution of this work is to change weights of easy, medium, and hard training subsets.**
>
> We presented approaches to (a) encode prior knowledge about sample difficulty into CL paradigms and (b) discover curricula from data (which are the two contributions of this work). Our framework encompasses several existing CL approaches; provides insights about data from the learned curricula for *easy*, *medium*, and *hard* samples; and illustrates major differences among curricula learned for different datasets.
>
> **What is the result if samples are randomly assigned to different difficulty classes? Does difficulty play an important role?**
>
> We thank the reviewer for suggesting this experiment. We have done this analysis with the SNLI balanced dataset by randomly permuting the difficulty classes of samples while conserving the original distribution of samples across difficulty classes. The following table shows the results, where each value is an average of 5 seeds. The specialized curricula are from Figures 5a (SBE) and 5b (SBL).
>
> | Method | Accuracy |
> | ------------ | ---- |
> | Loss (sp) | 76.40 (± 0.26) |
> | Loss (inc) | 75.89 (± 0.38) |
> | Ent (sp) | 73.45 (± 0.31) |
> | Ent (inc) | 72.87 (± 0.4) |
> | No-CL | 70.68 (± 0.53) |
> | Loss (sp) [permuted] | 66.81 (± 0.7) |
> | Loss (inc) [permuted] | 65.90 (± 0.58) |
> | Ent (sp) [permuted] | 68.70 (± 0.48) |
> | Ent (inc) [permuted] | 68.84 (± 0.88) |
>
> The results show that random assignment of difficulty leads to a significant reduction in the performance, which we attribute to the discrepancy in the difficulty of samples in each set.
>
> In addition to the above experiment, we previously measured the correlation between loss values of a trained model and the entropy values of samples. The resulting correlations are 0.28, 0.45, 0.40 for SNLI, Alcohol, and Cancer datasets respectively. A moderate correlation shows that entropy can provide good (but not perfect) estimates of sample difficulty to the model. The works of Nie et al. (EMNLP 2020) and Yang et al. (ACL 2019) show further evidence that human judgment can be used as a good measure of difficulty for models.
>
> **Can a different number of classes (less or more than three) be considered?**
>
> Yes, our method can work with any number of difficulty classes. One promising approach to obtain more than three difficulty classes is through clustering samples based on prior/general knowledge about linguistic difficulty in NLP, e.g., linguistic features like text length, type-token ratio, word frequency distribution, the range of surface and particular syntactic structures, and their general frequency of use, etc., can be used to cluster the samples into a desired number of difficulty classes. We note that most CL methods split their data into *easy* and *hard* difficulty groups. We chose three classes, adding *medium* because three classes provide finer-grained difficulty information and can filter out too easy or hard samples (Guo et al., ECCV’18).
>
> **Performance gains against other CL approaches are small, considering error bars in Figure 4.**
>
> We note that the errors bars represent the 95% confidence interval. The average accuracy of our model and the best performing CL baseline on SNLI is 77.50 vs 73.02. The corresponding performance on Cancer and Alcohol datasets are 72.99 vs 71.48, and 77.25 vs 77.10, respectively. The p-values of SNLI and Cancer are smaller than 0.05.
>
> **Model-level curriculum learning approaches should be acknowledged.**
>
> We updated the background section of our submission by citing several papers related to model-level and task-level CL, based on the survey paper that the reviewer suggested, and highlighted their difference with data-level CL (paragraph 2, Introduction).
>
> **Clarification about sample weighting approach in the introduction.**
>
> We updated the introduction with a concrete description of the sample weighting approach (paragraph 4, Introduction)

---

> > ### Comment · Reviewer_mHjR · 2021-11-21
> > **Reply to rebuttal**
> >
> > The rebuttal addressed some of the weakness, but not the most important points (novelty, claims in the introduction). I will thus keep my recommendation unchanged.

---

> > > ### Author Response · Authors · 2021-11-22
> > > **Addressing Novelty and Introduction Claims**
> > >
> > > We thank the reviewer for their time and comments.
> > >
> > > **The contributions are not novel.**
> > >
> > > To our knowledge, there is no CL approach that can create and explore a curricula space for curricula discovery and model generalization. Our approach provides a solution to this problem. In contrast to most previous CL approaches that find a *single* curriculum that works best on a specific task and dataset, our approach demonstrates that there could be *several* fundamentally different curricula that work equally well on a specific task and dataset (as shown in Appendix D). We believe the above contributions are novel and significant to CL.
> > >
> > > **Claim in the introduction that weights are predicted by the model.**
> > >
> > > In the revision on Nov 15, we added the following sentence to the introduction: "Each weight function is controlled by two parameters, which can be set empirically or adjusted using Bayesian optimization." This sentence clarifies that the weights are not predicted by the model, but adjusted through hyperparameter tuning.

---

### Official Review · Reviewer_gK7s · 2021-11-01

**Correctness:** 3
**Technical Novelty And Significance:** 2
**Empirical Novelty And Significance:** 2
**Recommendation:** 3
**Confidence:** 4

**Main Review:**

In general, the approach appears to be rather constrained (only 3 difficulty levels; limited to sigmoids; reliance in part on multi-annotator labels), may be onerous to implement (pre-training a baseline; fitting the sigmoids; making manual decisions on entropy vs loss) and was validated on rather small data (balanced datasets with <10k samples) to be useful in practice.

Strengths:
1. Addresses a hard unsolved problem.
2. Modeling simplicity of the curriculum functional class (simple sigmoids that would cover most of the human-hypothesized heuristics; however, it may not be what we need - see below).

Weaknesses:
1. I find the approach conceptually different to classical curriculum learning, since it reweighs instances in the loss rather than schedules them in a particular order (except for the zero-weight case, equivalent to example skipping, where both ways converge); as far as could tell, SGD still samples data uniformly at any time. Besides, since the loss function changes, the baseline and the weighted tasks optimize different objectives and it's not straight-forward to compare them.
2. Regarding sigmoids, I'm not convinced that practical cases of ML curricula can be covered by such simple monotonic schedules; rather, the functional class should be capable of dynamically changing the scheduling weight depending on the learning stage to mitigate forgetting and revisiting learned difficulty levels (and this encompasses monotonic functions too). In fact, such bandit-based curricula increasing/decreasing importance dynamically have been already proposed (Graves et al, 2017, https://arxiv.org/abs/1704.03003) and even evaluated for NLP tasks beyond classification (Kreutzer et al, 2021, https://arxiv.org/abs/2110.06997). The draft could be completed with a discussion of why the approach of Graves et al. is not considered or, better, a comparison experiment could be added. Note that the bandit approach is free from limitations such as number of difficulty levels.
3. The requirement to pre-train the baseline prior to curriculum-enabled training defeats one of the main purposes of curriculum learning -- saving resources in the large data regime and relieving developer from manual training design. Again, dynamic curricula that are trained on the fly would be a more practical approach, as they don't require pre-training.
4. The entropy-based curriculum relies heavy on assumed human-perceived difficulty, which may not be the same for the network's point of view. The fact, that Zhang et al. (2018) and Kocmi & Bojar 2017 both found that reverse curriculum works similarly well, witnesses that human intuitions of what is difficult for the network may be wrong.
5. It looks like data balancing increases the gap to the no-curriculum approach. This again changes the task (in addition to different objectives) and raises the question of practical relevance for tasks where such balancing is not desired or not possible (seq2seq).

Minor:
- i found Fig. 1 redundant, it is just illustrating what is sigmoid function
- fuzzy wording in a few places ("good metric", "fairly distributed", "significant variance")
- change \citet to \citep in some places, Richards (1959), Shannon (2001)
- sec 3.2: "configurations is" -> "configuration is"
- "in Nie et al (2020), which" -> "by Nie et al (2020), who"

**Summary Of The Paper:**

The paper proposes to learn training curricula by considering 3 sigmoids (representing "easy", "mid" and "hard" difficulty levels), that would define instances' weight as a function of time. The sigmoids' parameters are fitted to 3-bucket values of entropy of multi-annotator labels (that need to pre-exist), or of the instances' loss value dynamics (as provided by a baseline, no-curriculum, model that also needs to be trained). The paper emphasizes that the approach can replicate some of existing curriculum heuristics (e.g. easy-to-hard, hard-to-easy etc.), which however so far failed to produce a general recipe of training deep models, -- this diminishes the value of this flexibility. Finally, the approach is illustrated on 3 small to mid-size datasets with largest gaps to no-curriculum achieved on reduced datasets to re-balance classes.



**Summary Of The Review:**

In my opinion, the proposed approach is too limited to be of practical usefulness, and its weaknesses outweigh the strengths so that the current draft is below the ICLR bar.

---

> ### Author Response · Authors · 2021-11-15
> **Authors' Response to Reviewer gK7s**
>
> We thank the reviewer for their time and comments. The following is a response to comments raised by the reviewer.
>
> **Comparison against bandit-based CL (Graves et al., ICML 2017).**
>
> We considered state-of-the-art baselines that do not require an extra memory of O(*n*), *n* = number of samples, for storing additional information about sample weights and model during training. Methods such as bandit-based CL (Graves et al., ICML 2017) as well as more recent approaches such as Dynamic Instance Hardness (Zhou et al., NeurIPS 2020) and Data Parameters (Saxena et al., NeurIPS 2019) require extra memory. In addition, our approach enables encoding prior knowledge about sample difficulty into CL and falls under instructor-student collaborative approaches, while bandit-based and DIH methods are student-driven and don't have a systematic approach to use prior knowledge for CL. The paper has been updated to highlight these differences with the above models (paragraph 2 in the Introduction).
>
> **Practical cases of ML curricula may not be covered by simple monotonic schedules.**
>
> The focus and novelty of the proposed approach are in finding curriculum configurations for given scheduling functions, i.e. curriculum discovery. We chose generalized logistic functions as they present a system for ordering samples of different difficulties with a wide range of ordering strategies and relative weighting. We agree with the reviewer that a *monotonic* function may not be optimal in terms of coverage, however, generalized logistic functions do dynamically change the scheduling weights with respect to the learning stage and can mitigate forgetting by increasing weights. The proposed method can work with any other weighting function.
>
> **Since loss function changes during training, the baseline, and the weighted tasks optimize different objectives and it's not straightforward to compare them.**
>
> It is possible to compare CL models from the data viewpoint, where CL strategies differ in the order and frequency of presenting data instances. The weighted objective is equivalent to the original objective with some instances appearing multiple times (those with higher weights) or less frequently (those with lower weights) during training. The resulting performance can be directly compared as all models use the same data and criteria for training, validation, and testing.
>
> **Requirement for a full run (pre-training) defeats the purpose of CL.**
>
> Our approach employs entropy and average loss to estimate sample difficulty. Entropy information does not require pre-training as multi-annotation information is available for most NLP datasets at their creation time. Loss-based difficulty estimates are obtained through pre-training of a baseline model. However, pre-training (full run) is not a requirement— as discussed in Zhou et al (NeurIPS 2020), the exponential moving average of loss at early stages of training (e.g., the first few epochs) can be used to accurately estimate loss-based sample difficulty without pre-training.
>
> **Is there empirical evidence that annotator disagreement is a good measure of difficulty?**
>
> We measured the correlation between the loss values obtained from a trained model and the entropy values of samples. The resulting correlations are 0.28, 0.45, 0.40 for SNLI, Alcohol, and Cancer datasets respectively. A moderate correlation shows that entropy can provide good but not perfect estimates of sample difficulty to the model. The works of Nie et al. (EMNLP 2020) and Yang et al. (ACL 2019) show further evidence that human judgment can be used as a good measure of difficulty for models.
>
> **Model applicability on tasks where data balancing is not desired or not possible (seq2seq).**
>
> Data balancing is not a requirement of our model, although the model shows greater performance gains on balanced datasets. In terms of Full datasets, our approach outperforms other methods across easy and medium samples. No-CL is the best performing model on hard samples and the worst performing model on other samples. As Figure 9 shows, the overall accuracy of our approach is still better than No-CL and the other CL methods.

---

> > ### Comment · Reviewer_gK7s · 2021-11-21
> > **Some points are cleared up**
> >
> > Although few point are addressed by the authors, I keep my evaluation unchanged. It's a good first step but, as other reviewers note, the contribution would be more solid with evaluation on standard (and large) benchmarks where curricula are expected to have most important impact, comparisons to no-pretraining baselines and with more levels of difficulty, and results with significance scores.
> >
> > **Comparison against bandit-based CL (Graves et al., ICML 2017).**
> >
> > I would not agree that Graves et al. 17 requires additional memory worth mentioning -- normally, it would be tiny O(number of data groups) instead of O(samples). E.g., taking your setting and assuming same difficulty splits are given, its memory overhead would be just three (i.e hard, mid, easy) float numbers to store the Exp3 weights. Exp3 then would sample a group and proceed to a uniform sample from the group. Moreover, in practice data often comes naturally split into either domains or tasks or some other subsets, so we might even dispense with additional difficulty splits and could use the available grouping.
> >
> > **Practical cases of ML curricula may not be covered by simple monotonic schedules.**
> >
> > I don't quite see how the method would mitigate forgetting of a class $c$ (e.g. that has been assigned a decreasing sigmoid as the schedule). As far as I understand the $r_c$ and $s_c$ are fixed for whole CL run and cannot dynamically change.
> >
> > **The baseline, and the weighted tasks optimize different objectives and it's not straightforward to compare them**
> >
> > Yes, one could just compare according to the final test metrics and disregard the learning setup, but then comparisons of the loss dynamics is not very illuminating since it refers to completely different tasks (incl. drawing conclusions from Fig. 3 to guide the proposed approach). On more theoretical note, and to cite Graves et al. 17 again, optimizing online regret one does have a connection to the original batch loss: i.e. success in the former leads to success in the latter (see Bottou et al 16, https://arxiv.org/abs/1606.04838, p. 38).
> >
> > **Requirement for a full run (pre-training) defeats the purpose of CL.**
> >
> > Thank you for the reference Zhou et al. First few epochs may be provide some insights, although it becomes a yet another hyper-parameter.
> >
> > **Is there empirical evidence that annotator disagreement is a good measure of difficulty?**
> >
> > Thank you.
> >
> > **Model applicability on tasks where data balancing is not desired or not possible (seq2seq).**
> >
> > Here I would side with the reviewer XfAD in the skepticism regarding the significance of results like in Fig 9. Without those it's hard to be confident that "the accuracy is better" and to draw ranking conclusions. Would it it possible to perform significance testing and add the p-values?

---

> > > ### Author Response · Authors · 2021-11-22
> > > **Table 1: Results of Statistical Significance Test**
> > >
> > > | Dataset | Method (1) | Method (2) | Accuracy (1) | Accuracy (2) | p-value | Stars |
> > > | ---------- | ------- | ------- | ------- | ------- | ------- | ------- |
> > > | snli_balanced | Loss (sp) | MentorNet | 77.4 | 73.0 | 0 | *** |
> > > |  |  | DP | 77.4 | 72.8 | 0 | *** |
> > > |  |  | SL | 77.4 | 72.3 | 0 | *** |
> > > |  |  | No-CL | 77.4 | 72.2 | 0 | *** |
> > > |  | Ent (sp) | MentorNet | 74.9 | 73.0 | 0 | * |
> > > |  |  | DP | 74.9 | 72.8 | 0 | ** |
> > > |  |  | SL | 74.9 | 72.3 | 0 | *** |
> > > |  |  | No-CL | 74.9 | 72.2 | 0 | *** |
> > > |  | Loss (inc) | MentorNet | 76.9 | 73.0 | 0 | *** |
> > > |  |  | DP | 76.9 | 72.8 | 0 | *** |
> > > |  |  | SL | 76.9 | 72.3 | 0 | *** |
> > > |  |  | No-CL | 76.9 | 72.2 | 0 | *** |
> > > |  | Ent (inc) | MentorNet | 74.4 | 73.0 | 0 | * |
> > > |  |  | DP | 74.4 | 72.8 | 0 | ** |
> > > |  |  | SL | 74.4 | 72.3 | 0 | *** |
> > > |  |  | No-CL | 74.4 | 72.2 | 0 | *** |
> > > |  | No-CL | MentorNet | 72.2 | 73.0 | 0.08 |  |
> > > |  |  | DP | 72.2 | 72.8 | 0.09 |  |
> > > |  |  | SL | 72.2 | 72.3 | 0.82 |  |
> > > | alcohol_balanced | Loss (sp) | DP | 77.2 | 77.1 | 0.72 |  |
> > > |  |  | MentorNet | 77.2 | 76.9 | 0.25 |  |
> > > |  |  | SL | 77.2 | 76.6 | 0.19 |  |
> > > |  |  | No-CL | 77.2 | 75.7 | 0.07 |  |
> > > |  | Ent (sp) | DP | 76.7 | 77.1 | 0.53 |  |
> > > |  |  | MentorNet | 76.7 | 76.9 | 0.74 |  |
> > > |  |  | SL | 76.7 | 76.6 | 0.83 |  |
> > > |  |  | No-CL | 76.7 | 75.7 | 0.24 |  |
> > > |  | Loss (inc) | DP | 76.1 | 77.1 | 0.17 |  |
> > > |  |  | MentorNet | 76.1 | 76.9 | 0.22 |  |
> > > |  |  | SL | 76.1 | 76.6 | 0.53 |  |
> > > |  |  | No-CL | 76.1 | 75.7 | 0.59 |  |
> > > |  | Ent (inc) | DP | 77.0 | 77.1 | 0.81 |  |
> > > |  |  | MentorNet | 77.0 | 76.9 | 0.52 |  |
> > > |  |  | SL | 77.0 | 76.6 | 0.31 |  |
> > > |  |  | No-CL | 77.0 | 75.7 | 0.11 |  |
> > > |  | No-CL | DP | 75.7 | 77.1 | 0.09 |  |
> > > |  |  | MentorNet | 75.7 | 76.9 | 0.14 |  |
> > > |  |  | SL | 75.7 | 76.6 | 0.28 |  |
> > > | cancer_balanced | Loss (sp) | DP | 72.9 | 71.4 | 0.06 |  |
> > > |  |  | MentorNet | 72.9 | 71.1 | 0.04 | * |
> > > |  |  | No-CL | 72.9 | 70.7 | 0 | * |
> > > |  |  | SL | 72.9 | 70.6 | 0.01 | * |
> > > |  | Ent (sp) | DP | 72.4 | 71.4 | 0.13 |  |
> > > |  |  | MentorNet | 72.4 | 71.1 | 0.08 |  |
> > > |  |  | No-CL | 72.4 | 70.7 | 0.01 | * |
> > > |  |  | SL | 72.4 | 70.6 | 0.02 | * |
> > > |  | Loss (inc) | DP | 71.7 | 71.4 | 0.76 |  |
> > > |  |  | MentorNet | 71.7 | 71.1 | 0.51 |  |
> > > |  |  | No-CL | 71.7 | 70.7 | 0.23 |  |
> > > |  |  | SL | 71.7 | 70.6 | 0.23 |  |
> > > |  | Ent (inc) | DP | 71.0 | 71.4 | 0.34 |  |
> > > |  |  | MentorNet | 71.0 | 71.1 | 0.88 |  |
> > > |  |  | No-CL | 71.0 | 70.7 | 0.42 |  |
> > > |  |  | SL | 71.0 | 70.6 | 0.42 |  |
> > > |  | No-CL | DP | 70.7 | 71.4 | 0.14 |  |
> > > |  |  | MentorNet | 70.7 | 71.1 | 0.52 |  |
> > > |  |  | SL | 70.7 | 70.6 | 0.83 |  |
> > > | snli_special | Loss (sp) | DP | 86.0 | 86.0 | 0.88 |  |
> > > |  |  | SL | 86.0 | 85.8 | 0.38 |  |
> > > |  |  | MentorNet | 86.0 | 85.8 | 0.22 |  |
> > > |  |  | No-CL | 86.0 | 85.6 | 0.23 |  |
> > > |  | Ent (sp) | DP | 85.9 | 86.0 | 0.60 |  |
> > > |  |  | SL | 85.9 | 85.8 | 0.71 |  |
> > > |  |  | MentorNet | 85.9 | 85.8 | 0.47 |  |
> > > |  |  | No-CL | 85.9 | 85.6 | 0.37 |  |
> > > |  | Loss (inc) | DP | 86.0 | 86.0 | 0.90 |  |
> > > |  |  | SL | 86.0 | 85.8 | 0.55 |  |
> > > |  |  | MentorNet | 86.0 | 85.8 | 0.44 |  |
> > > |  |  | No-CL | 86.0 | 85.6 | 0.34 |  |
> > > |  | Ent (inc) | DP | 85.9 | 86.0 | 0.85 |  |
> > > |  |  | SL | 85.9 | 85.8 | 0.56 |  |
> > > |  |  | MentorNet | 85.9 | 85.8 | 0.38 |  |
> > > |  |  | No-CL | 85.9 | 85.6 | 0.31 |  |
> > > |  | No-CL | DP | 85.6 | 86.0 | 0.21 |  |
> > > |  |  | SL | 85.6 | 85.8 | 0.47 |  |
> > > |  |  | MentorNet | 85.6 | 85.8 | 0.62 |  |
> > > | alcohol | Loss (sp) | SL | 78.7 | 78.8 | 0.60 |  |
> > > |  |  | DP | 78.7 | 78.8 | 0.54 |  |
> > > |  |  | MentorNet | 78.7 | 78.6 | 0.81 |  |
> > > |  |  | No-CL | 78.7 | 78.3 | 0.32 |  |
> > > |  | Ent (sp) | SL | 79.4 | 78.8 | 0.09 |  |
> > > |  |  | DP | 79.4 | 78.8 | 0.01 | * |
> > > |  |  | MentorNet | 79.4 | 78.6 | 0.07 |  |
> > > |  |  | No-CL | 79.4 | 78.3 | 0.02 | * |
> > > |  | Loss (inc) | SL | 78.8 | 78.8 | 0.94 |  |
> > > |  |  | DP | 78.8 | 78.8 | 0.96 |  |
> > > |  |  | MentorNet | 78.8 | 78.6 | 0.62 |  |
> > > |  |  | No-CL | 78.8 | 78.3 | 0.26 |  |
> > > |  | Ent (inc) | SL | 78.4 | 78.8 | 0.19 |  |
> > > |  |  | DP | 78.4 | 78.8 | 0.12 |  |
> > > |  |  | MentorNet | 78.4 | 78.6 | 0.57 |  |
> > > |  |  | No-CL | 78.4 | 78.3 | 0.82 |  |
> > > |  | No-CL | SL | 78.3 | 78.8 | 0.23 |  |
> > > |  |  | DP | 78.3 | 78.8 | 0.20 |  |
> > > |  |  | MentorNet | 78.3 | 78.6 | 0.52 |  |
> > > | cancer | Loss (sp) | No-CL | 73.0 | 72.7 | 0.37 |  |
> > > |  |  | DP | 73.0 | 72.5 | 0.29 |  |
> > > |  |  | SL | 73.0 | 71.7 | 0 | ** |
> > > |  |  | MentorNet | 73.0 | 71.5 | 0.01 | * |
> > > |  | Ent (sp) | No-CL | 72.3 | 72.7 | 0.56 |  |
> > > |  |  | DP | 72.3 | 72.5 | 0.79 |  |
> > > |  |  | SL | 72.3 | 71.7 | 0.17 |  |
> > > |  |  | MentorNet | 72.3 | 71.5 | 0.17 |  |
> > > |  | Loss (inc) | No-CL | 72.3 | 72.7 | 0.57 |  |
> > > |  |  | DP | 72.3 | 72.5 | 0.75 |  |
> > > |  |  | SL | 72.3 | 71.7 | 0.36 |  |
> > > |  |  | MentorNet | 72.3 | 71.5 | 0.29 |  |
> > > |  | Ent (inc) | No-CL | 72.9 | 72.7 | 0.66 |  |
> > > |  |  | DP | 72.9 | 72.5 | 0.53 |  |
> > > |  |  | SL | 72.9 | 71.7 | 0.05 |  |
> > > |  |  | MentorNet | 72.9 | 71.5 | 0.06 |  |
> > > |  | No-CL | DP | 72.7 | 72.5 | 0.75 |  |
> > > |  |  | SL | 72.7 | 71.7 | 0.01 | * |
> > > |  |  | MentorNet | 72.7 | 71.5 | 0.04 | * |

---

> > > > ### Comment · Reviewer_gK7s · 2021-11-30
> > > > **Thanks**
> > > >
> > > > Thank you for the p-values. They suggest that the approach worked across its variations only for one data dataset and only for the balanced version of it, which is, as discussed before, an operation not always possible, desirable or allowed to do.
> > > >
> > > > Regarding the rest of comments, they may make sense in principle, but eventually should be supported by numbers, in addition to intuitions.
> > > >
> > > > In summary, I think the paper improved with the modifications (esp. p-values), and while being on the right track, it does not make it to the ICLR bar quite yet. I'm keeping my score unchanged (looking forward to see this approach appear with comparisons on large data and against the mentioned baselines).

---

> > > ### Author Response · Authors · 2021-11-23
> > > **Further Clarification and p-values**
> > >
> > > We thank the reviewer for their time and comments.
> > >
> > > **Evaluation of standard (and large) benchmarks is needed.**
> > >
> > > We understand the reviewer's goal is to provide a solution to better illustrate the impact of our work, and we thank the reviewer for the comment. However, we evaluated our work on SNLI, which is a large benchmark dataset in NLP. In addition, recent work shows that CL models can improve performance in cases of limited or noisy data (Wu et al., ICLR 2021).
> > >
> > > **Comparison against bandit-based CL (Graves et al., ICML 2017).**
> > >
> > > The reviewer is correct about the memory requirement of the bandit-based CL approach, which is small — O(number of data groups). However, there are still major characteristic differences between bandit-based CL, which selects a single task at every step 1...T to maximize the reward (loss or prediction gain), and our approach which learns all tasks (difficulty groups) jointly by weighting them differently throughout training. In addition, we compared our approach against more recent and state-of-the-art CL methods.
> > >
> > > **The baseline and the weighted tasks optimize different objectives and it's not straightforward to compare them**
> > >
> > > The objectives optimized by our approach and baselines are not completely different. They all operate under the premise of instance loss and confidence, where the weighted batch loss is controlled by instance losses and confidence values (which can increase or decrease at subsequent iterations). In this respect, the (weighted) loss dynamics of different CL methods are comparable.
> > >
> > > **Forgetting matters, practical cases of ML curricula may not be covered by simple monotonic schedules.**
> > >
> > > The forgetting challenge is discussed in the second paragraph of our conclusion section.
> > > > ... This challenge can be addressed in future work by adding an extra set of logistic functions and having every weight function be a linear combination of two or more logistic functions, which makes it possible to represent non-monotonic functions. Nevertheless, we have demonstrated that major CL approaches that estimate difficulty as a function of training loss result in monotonic curves because of gradient descent that causes training loss to be non-increasing.
> > >
> > > In addition, our model with logistic functions can partially mitigate forgetting through its shift parameter*s*, which allows the curriculum to assign a non-zero weight to a class until the end of training if necessary.
> > >
> > > **Significance testing and p-values**
> > >
> > > Table 1 shows p-values obtained by comparing our approach and the baselines. We also include p-values comparing No-CL against baselines. The star column shows statistically significant differences at p<0.0005 (***) and p<0.05 (*); p-values are based on a two-tailed paired t-Test with 5 seeds. The results illustrate that our approach performs comparably or better than competing baselines. In addition, significance testing against No-CL shows that our approach is the only one that shows significantly better performance than No-CL. We believe works that present "correct" and "novel" approaches that show comparable performance to that of current models are presentable research.

---

### Official Review · Reviewer_XfAD · 2021-11-02

**Correctness:** 3
**Technical Novelty And Significance:** 2
**Empirical Novelty And Significance:** 2
**Recommendation:** 6
**Confidence:** 4

**Main Review:**

#### Strengths
- Proposed curriculum method encompasses other well known curricula by parameterizing the data partition and weighing schemes.
- Data partitioning is effective, on the three datasets the authors evaluate upon, as is evident from improvements over no curriculum approach.
- Specialized curricula obtained using curriculum discovery improves over no curriculum and other state of the art approaches.
- Sample weight patterns shows the relative importance of samples on different datasets, which is interesting.
- One quite interesting advantage of this approach is that curriculum discovered for smaller, balanced datasets work well on larger datasets.

#### Weaknesses
- The proposed method partitions a dataset into three classes $\textit{easy}$, $\textit{medium}$ and $\textit{hard}$. Will this partitioning scheme work for different types of datasets, where there might be multiple levels of difficulty, multiple sub tasks? How will this method scale?
- Using human agreement as a measure of entropy might not be applicable to datasets where there is less ambiguity between samples. That will cause majority of samples to be in a particular partition. For those datasets using entropy as a measure to partition samples might not be optimal.
-  In Section 2.3, the authors describe how the proposed framework encompasses other well known CL approaches. It will be interesting to add more analysis on how often the TPE algorithm(Section 2.4) discovers curriculums where the parameters fall into pruning or sub-sampling strategy space.
- How does this approach scale to larger datasets in vision like Imagenet, CIFAR etc, that other SOTA methods like SuperLoss evaluate on? Authors evaluate on three datasets, not all of which are well known to the community. I will encourage the authors to also evaluate this approach on more popular datasets.
- How significant are the improvements over other baselines? Showing average over full and balanced datasets(Fig 4) hides some important issues. It seems from Figure 10, that DP method outperforms the proposed CL method on full datasets. We should pay more interest to this number as we would want any approach to work well on full dataset, rather than a subset(balanced) dataset.
- Hard examples are down weighted more aggressively in this approach. It can be seen that the proposed approach often lags behind other approaches like No-CL, MentorNet, DP when it comes to hard examples. This issue is more prominent when it comes to full datasets. Authors should provide some additional analysis on these hard samples from these datasets for more clarity. What % of these datasets are partitioned into $\textit{hard}$ class?

**Summary Of The Paper:**

This paper proposes a new parameterized data partitioning and weighing scheme, that partitions data into three groups {easy, medium, hard} and determines a curriculum based on relative importance of different samples. They evaluate on three datasets (full and balanced versions) and show improvements over other CL approaches. The curriculum also provides interesting insights about the datasets and scale from smaller datasets to larger datasets.

**Summary Of The Review:**

I like the idea of encompassing different curriculum approaches inside a parameterized function. The authors show improvements over other approaches on three datasets with curriculum discovered by this approach. My reservations are mostly around how well this approach will scale to larger datasets and on unbalanced datasets. Some results show that other methods are better when it comes to full datasets. Due to these reasons, I will am recommending the current score. I encourage authors to address these concerns and I will be happy to bump up my score.

---

> ### Author Response · Authors · 2021-11-15
> **Author's Response to Reviewer XfAD**
>
> We thank the reviewer for their time and comments. The following is a response to comments raised by the reviewer.
>
> **Will partitioning into easy/med/hard samples work with datasets with multiple levels of difficulty?**
>
> Most CL methods split their data into *easy* and *hard* difficulty classes. However, we partitioned our datasets into three difficulty classes, adding *medium*, to provide finer-grained difficulty information (Guo et al., *ECCV*. 2018). If difficulty estimates are available (please see below), our method can work with any number of difficulty classes without modification.
>
> **Partitioning based on entropy might not be applicable to datasets where there is less ambiguity between samples.**
>
> Due to the complexity and intrinsic ambiguity of human language, annotator disagreements are prevalent in most NLP datasets, and therefore it is fair to expect a reasonable data distribution across difficulty classes. However, in case of less/no ambiguity between samples (i.e. existence of a large dominant difficulty class), the dominant class may be further clustered into several sub-classes of difficulty, which will be parameterized and weighed differently by our model. The clustering, however, should be performed based on some prior/general knowledge about sample difficulty. For example, linguistic measures such as text length, type-token ratio, word frequency distribution, the range of surface and particular syntactic structures and their general frequency of use, etc., can be used to cluster the samples into any desired number of difficulty classes.
>
> **Can the approach scale to [larger] datasets like CIFAR or Imagenet?**
>
> Our SNLI dataset consists of 39k training samples, which is comparable in size to CIFAR (~50k training samples). Our criterion for choosing Alcohol and Cancer datasets was the availability of multiple annotations per instance in these datasets. Our method can scale to larger datasets because: (1) hyperparameter search can scale to large datasets: as demonstrated in the paper, curricula discovered for smaller datasets work well on larger datasets. So, it is sufficient to perform the search on a subset of the dataset and apply the discovered curricula to the larger datasets. (2) loss-based difficulty estimates do not necessarily require full training of a baseline model: as shown in (Zhou et al, NeurIPS 2020), the exponential moving average of loss at early stages of training (e.g., the first few epochs) is an effective metric to estimate sample difficulty. Therefore, a warm-up training period is sufficient and full training is not required.
>
> **Further analysis on down-weighting of hard samples.**
>
> The proportions of easy, medium and hard samples in our three datasets are as follows:
>
> | Dataset | Easy | Medium | Hard |
> | ----- | -------- | -------- | -------- |
> | SNLI | 84% | 14% | 2% |
> | Alcohol | 55% | 29% | 16% |
> | Cancer | 36% | 39% | 25% |
>
> The hyperparameter search algorithm explores the region of curriculum space where hard samples have a high weight (positive r in Eq (1)). However, the best-performing curricula are those that assign lower weights to harder samples. In addition, the outcome of our approach supports the hypothesis that instances with high disagreement are likely to be noisy. Such samples are referred to in Bengio et al. (ICML 2019) as "falling near the decision surface" (or on the incorrect side).

---

> > ### Comment · Reviewer_XfAD · 2021-11-23
> > **Thanks for the response**
> >
> > I thank the authors for the response and addressing some of the queries I had. I will maintain my current rating, as some of the reservations I have around the significance of the results(results don't always outperform the baselines on full dataset) and applicability of this method to larger/popular datasets need more details.

---

### Author Response · Authors · 2021-11-22
**Summary of Modifications**

We again thank the reviewers for their valuable comments that have helped us improve the draft. The following is a summary of the modifications to the manuscript.

**Nov 15**

- Clarification about the sample weighting approach in the introduction.
- Recognize model-based and task-based CL in the review of the related work.
- Discuss differences with Automated Curriculum Learning (Graves et al., ICML 2017).

 **Nov 22**

- Revised the text throughout the paper to improve presentation.
- Removed the footnote on page 6 about the Cancer dataset.
- Removed bandit-based CL from the list of methods requiring O(n) parameters.

---

### Decision · Program_Chairs · 2022-01-20

**Decision:**

Reject

**Comment:**

The paper proposes a new curriculum learning framework by parameterizing data partitioning and weighting schemes. Extensive experiments are performed on three different datasets to demonstrate the effectiveness of the proposed framework. The reviewers acknowledged that the proposed framework is interesting as it encompasses several existing curriculum learning methods. However, the reviewers pointed out several weaknesses in the paper and shared concerns, including the scalability of the framework to larger datasets and the significance of the improvements over baselines. I want to thank the authors for their detailed responses. Based on the reviewers’ concerns and follow-up discussions, there was a consensus that the work is not ready for publication. The reviewers have provided detailed feedback to the authors. We hope that the authors can incorporate this feedback when preparing future revisions of the paper.